



# Modeled and observed properties related to the direct aerosol radiative effect of biomass burning aerosol over the Southeast Atlantic

Sarah J. Doherty[1,2], Pablo E. Saide[3,4], Paquita Zuidema[5], Yohei Shinozuka[6,7], Gonzalo A. Ferrada[8], Hamish Gordon[9], Marc Mallet[10], Kerry Meyer[11], David Painemal[12,13], Steven G. Howell[14], Steffen Freitag[14], Amie Dobracki[5], James R. Podolske[7], Sharon P. Burton[13], Richard A. Ferrare[13], Calvin Howes[3], Pierre Nabat[10], Gregory R. Carmichael[8], Arlindo da Silva[15], Kristina Pistone[6,7], Ian Chang[16], Lan Gao[16], Robert Wood[2] and Jens Redemann[16]

[1] Cooperative Institute for Climate, Ocean and Ecosystem Studies, Seattle, WA, USA
[2] Department of Atmospheric Science, University of Washington, Seattle, WA, USA
[3] Department of Atmospheric and Oceanic Sciences, University of California, Los Angeles, CA USA
[4] Institute of the Environment and Sustainability, University of California, Los Angeles, CA, USA
[5] Rosenstiel School of Marine and Atmospheric Science, University of Miami, Miami, FL, USA
[6] Bay Area Environmental Research Institute, Moffett Field, CA, USA
[7] NASA Ames Research Center, Moffett Field, CA, USA
[8] Center for Global and Regional Environmental Research, The University of Iowa, Iowa City, IA, USA
[9] Engineering Research Accelerator and Center for Atmospheric Particle Studies, Carnegie Mellon University, Pittsburgh, PA, USA
[10] CNRM, Université de Toulouse, Météo-France, CNRS, Toulouse, France
[11] NASA Goddard Space Flight Center, Greenbelt, Maryland, MD 20771, USA
[12] Science Systems and Applications Inc., Hampton, Virginia 23666, USA
[13] NASA Langley Research Center, Hampton, Virginia 23691 USA
[14] University of Hawaii at Manoa, Honolulu, HI, USA
[15] Global Modeling and Assimilation Office, NASA Goddard Space Flight Center, Greenbelt, MD, USA
[16] School of Meteorology, University of Oklahoma, Norman, OK, USA

*Correspondence to*: Sarah J. Doherty (sarahd@atmos.washington.edu)

**Abstract.** Biomass burning smoke is advected over the southeast Atlantic Ocean between July and October of each year. This smoke plume overlies and mixes into a region of persistent low marine clouds. Model calculations of climate forcing by this plume vary significantly, in both magnitude and sign. The NASA EVS-2 (Earth Venture Suborbital-2) ORACLES (ObseRvations of Aerosols above CLouds and their intEractionS) project deployed for field campaigns off the west coast of Africa in three consecutive years (Sept., 2016; Aug., 2017; and Oct., 2018) with the goal of better characterizing this plume as a function of the monthly evolution, by measuring the parameters necessary to calculate the direct aerosol radiative effect. Here, this dataset and satellite retrievals of cloud properties are used to test the representation of the smoke plume and the underlying cloud layer in two regional models (WRF-CAM5 and CNRM-ALADIN) and two global models (GEOS and UM-UKCA). The focus is on comparisons of those aerosol and cloud properties that are the primary determinants of the direct aerosol radiative effect, and on the vertical distribution of the plume and its properties. The representativeness of the observations to monthly averages are tested for each field campaign, with the sampled mean aerosol light extinction



generally found to be within 20% of the monthly mean at plume altitudes. When compared to the observations, in all models the simulated plume is too vertically diffuse, has smaller vertical gradients, and, in two of the models (GEOS and UM-UKCA), the plume core is displaced lower than in the observations. Plume carbon monoxide, black carbon, and organic aerosol masses indicate under-estimates in modeled plume concentrations, leading in general to under-estimates in mid-visible aerosol extinction and optical depth. Biases in mid-visible single scatter albedo are both positive and negative across

the models. Observed vertical gradients in single scatter albedo are not captured by the models, but the models do capture the coarse temporal evolution, correctly simulating higher values in October (2018) than in August (2018) and September (2017). Uncertainties in the measured absorption Ångstrom exponent were large but propagate into a negligible (<4%) uncertainty in integrated solar absorption by the aerosol and therefore in the aerosol direct radiative effect. Model biases in cloud fraction, and therefore the scene albedo below the plume, vary significantly across the four models. The optical

thickness of clouds is, on average, well simulated in the WRF-CAM5 and ALADIN models in the stratocumulus region and is under-estimated in the GEOS model; UM-UKCA simulates significantly too-high cloud optical thickness. Overall, the study demonstrates the utility of repeated, semi-random sampling across multiple years that can give insights into model biases and how these biases affect modeled climate forcing. The combined impact of these aerosol and cloud biases on the direct aerosol radiative effect (DARE) is estimated using a first-order approximation for a sub-set of five comparison

gridboxes. A significant finding is that the observed gridbox-average aerosol and cloud properties yield a positive (warming) aerosol direct radiative effect for all five gridboxes, whereas DARE using the gridbox-averaged modeled properties ranges from much larger positive values to small, negative values. It is shown quantitatively how model biases can offset each other, so that model improvements that reduce biases in only one property (e.g., single scatter albedo, but not cloud fraction) would lead to even greater biases in DARE. Across the models, biases in aerosol extinction and in cloud fraction and optical

depth contribute the largest biases in DARE, with aerosol single scatter albedo also making a significant contribution.

## 1 Introduction

Climate forcing by both direct aerosol-radiation interactions and aerosol-cloud interactions is, in general, dependent on the vertical location of the aerosol relative to clouds, and especially so for absorbing aerosol (e.g., Samset et al., 2013). In the Southeast Atlantic region this is particularly true. From August through October there is a spatially broad, high-concentration

smoke plume that overlies, and in places and times mixes with, a persistent boundary layer cloud deck. As such, direct radiative forcing in the region is a strong function not only of smoke concentration, composition and vertical distribution, but also of the albedo below the plume. Over the SE Atlantic, this albedo is arguably driven primarily by cloud fraction and liquid water path, as well as by cloud droplet number concentration, with the latter controlled by aerosol-cloud microphysical interaction. Large-scale models have been shown to have large uncertainties and biases in their simulations of

both aerosol absorption (e.g., Sand et al., 2021; Brown et al., 2021) and low marine clouds (e.g., Noda and Sato, 2014; Kawai and Shige, 2020) in this region.



Modeled direct radiative forcing across the SE Atlantic has ranged from strongly negative to strongly positive, with much of this range determined by modeled cloud fraction (e.g., see Figure 2 of Zuidema et al., 2016 and Stier et al., 2013). In one

study, the direct aerosol radiative effect in the region changed from negative to positive as cloud fraction increased above 40% (Chand et al., 2009), assuming a mid-visible aerosol single scatter albedo (SSA) of 0.85 and for cloud optical depths averaging 7.8 (or cloud albedo of 0.5). For aerosol with lower SSA or for higher cloud albedo this transition would occur at a lower cloud fraction (Mallet et al., 2020). The sign and magnitude of the responses to this forcing (i.e. cloud adjustments formerly referred to as the "semi-direct effect") also depend strongly on the underlying cloud properties and the relative

vertical locations of the aerosol and cloud (e.g. Penner et al., 2003; Johnson et al., 2004; Sakaeda et al., 2011; Bond et al., 2013; Matus et al, 2015). Absorbing aerosol aloft has been linked to increased lower tropospheric stability and enhanced cloud cover and thickness, compared to cleaner environmental conditions. This has been attributed to heating of the air aloft, limiting the entrainment of dry air from the free troposphere into the marine boundary layer (Wilcox, 2010; Gordon et al., 2018), in turn enhancing low cloud cover (e.g., Johnson et al., 2004; Wilcox, 2010), with Herbert et al. (2020) indicating a

dependence on the cloud-aerosol layer distance. However, if the aerosol mixes into the clouds, atmospheric heating there may reduce cloud cover (Koch and Del Genio, 2010; Zhang and Zuidema, 2019).

The large uncertainty in aerosol climate forcing in the SE Atlantic was the impetus for the NASA ORACLES (ObseRvations of Aerosols above CLouds and their intEractionS) project, funded through the NASA Earth Venture Suborbital (EVS-2)

program (Redemann et al., 2021), as well as complementary campaigns (Zuidema et al., 2016; Formenti et al., 2019; Haywood et al., 2021). The ORACLES project explicitly measured aerosol properties necessary to calculate the direct aerosol radiative effect (DARE). The campaign included deployments of the NASA P-3 research aircraft to the SE Atlantic region based out of Walvis Bay, Namibia (27 August-27 September 2016) and São Tomé, São Tomé e Príncipe (09 August-02 September 2017 and 24 September-25 October 2018). The NASA ER-2 aircraft also deployed to Walvis Bay 26 August-

29 September 2016. The P-3 carried a suite of instruments to measure in-situ gas concentrations and aerosol microphysical, optical and chemical properties, to measure cloud microphysical properties, and to remotely sense both aerosols and clouds. It generally flew between 100m and 6km above the sea surface, capturing in-situ data in ramped or spiraling profiles, horizontal variations in level legs, and aerosol and trace gas columnar properties (e.g., aerosol optical depth, AOD) when flying below aerosol layers. The ER-2 flew at high altitude (19 km) and carried remote sensing instruments only, observing

both aerosols and clouds. (See Redemann et al., 2021 for a more complete overview of the ORACLES campaigns).

Here, aerosol and cloud properties observed during the three ORACLES deployment periods are compared to two regional models, the Weather Research and Forecasting model coupled with the physics package of Community Atmosphere Model (WRF-CAM5) and the Centre National de Recherches Aire Limitée Adaptation dynamique Développement InterNational

model (CNRM-ALADIN; hereafter simply ALADIN) and two global models, the Goddard Earth Observing System model





(GEOS) and the United Kingdom Chemistry and Aerosols Unified Model (UM-UKCA). Descriptions of each model are given below. The WRF-CAM5 and GEOS models are included because they were used for aerosol and meteorological forecasting during the ORACLES campaign. Similarly, the UM-UKCA modeling team participated in the U.K. CLARIFY (Cloud-Aerosol-Radiation Interactions and Forcing) campaign, which deployed out of Ascension Island (**Figure 1**) in 2017

(Haywood et al., 2021). In 2016 both ORACLES and the French AEROCLO-SA (AEROsol radiation and CLOuds in Southern Africa; Formenti et al., 2019) campaign deployed out of Walvis Bay, Namibia, with the latter focusing on near-coast aerosols. The ALADIN regional model version 6 used here (Mallet al., 2019, 2020; Nabat et al., 2020) simulated aerosol and clouds over the SE Atlantic as part of the AEROCLO-SA campaign.

**Figure 1.** The four transects used in the model-observation comparison are shown along with the P-3 (2016,

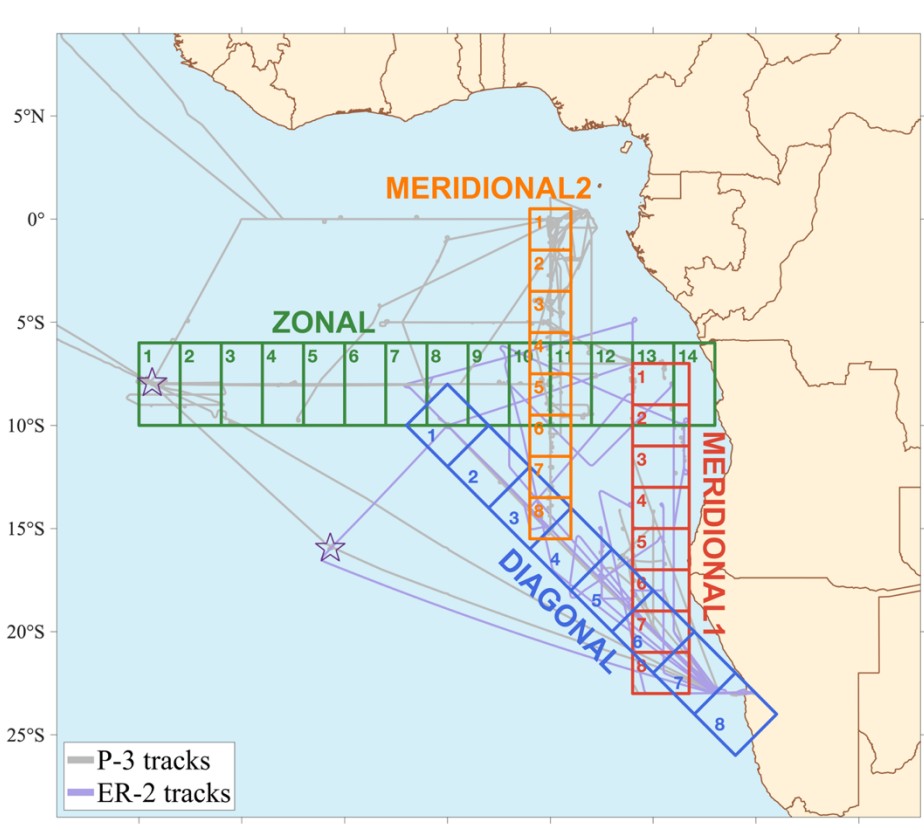

2017 and 2018) and ER-2 (2016) aircraft flight tracks. Gridboxes within each transect are numbered from north/west to south/east. See Supplementary Table S.1 for the gridbox coordinates. The locations of Ascension Island (8S, 14W) and St. Helena Island (16S, 6W) are shown as stars.

The properties included here (Section 2.1) allow for a first-order calculation of DARE (Section 6.0). Forcing through aerosol-cloud interactions is driven by a more complex set of processes – including the time-history of aerosols and clouds



(Diamond et al. 2018), aerosol physical and chemical properties (e.g., McFiggans et al., 2006; Kacarab et al., 2020), the micro- and macro-physical properties of the clouds (e.g. Stevens and Feingold, 2009; Koren et al., 2014; Gupta et al., 2020), and the thermodynamic state of the atmosphere. There is limited treatment herein of this broader set of variables, though testing model accuracy in representing cloud fraction does provide a critical zero-order step toward determining whether models might be capturing processes key to quantifying forcing through aerosol-cloud interactions.


This study builds on Shinozuka et al. (2020), which compared modeled and observed column aerosol properties for the September 2016 ORACLES deployment only. There, comparisons were presented as box-whisker plots of two-dimensional (2D; i.e. column) and three-dimensional (3D) variables. For 3D variables, values were binned into three discrete layers: the marine boundary layer, the free troposphere below 3 km, and 3-6 km altitude. The Shinozuka et al. (2020) study focused on

comparisons of layer-average carbon monoxide (CO) and aerosol properties, as well as the smoke layer bottom and top altitudes. Comparisons were made to six models: in addition to the four included herein, the EAM-E3SM and GEOS-Chem models provided statistics for the variables analyzed in Shinozuka et al. (2020).

## 2 Comparison overview

The comparisons presented here focus on vertically-resolved aerosol properties, where data are averaged into 500 m altitude

bins from the surface up to 6 km, and on the clouds below the biomass burning smoke plume. The observed and modeled aerosol and cloud properties are compared for multiple transects across the southeast Atlantic (**Figure 1**). Observed values of the aerosol properties are from the ORACLES research flights, while observed cloud properties are from satellite retrievals, since calculation of cloud fraction and optical depth from observations made from the aircraft are too limited to be of use in the statistical comparison presented here.

**2.1 Comparison variables**

The focus is on variables that are strongly related to direct aerosol radiative effect of the biomass burning aerosol. Vertically-resolved comparisons are made for the following variables measured in-situ from the NASA P-3, and measured with the high-spectral-resolution lidar, HSRL-2 ($\sigma_{ep}$ only), which deployed on the ER-2 in 2016 and the P-3 in 2017 and 2018:

- Carbon Monoxide (CO) mixing ratio

- Black Carbon (BC) concentration
- Organic Aerosol (OA) concentration
- Light extinction ($\sigma_{ep}$) [530nm from the in situ observations; 532nm from the HSRL-2; 550nm from the models]
- Single Scatter Albedo (SSA) [530nm from the observations; 550nm from the models]
- Aerosol Absorption Ångström Exponent (AAE) [440-670nm for the observations; 400-600nm for the WRF-CAM5,

GEOS and ALADIN models; 380-550nm for UM-UKCA]



- Aerosol Scattering Ångström Exponent (SAE) [440-670nm for the observations; 400-600nm for the WRF-CAM5, GEOS and ALADIN models; 380-550nm for UM-UKCA]
- Relative Humidity (RH)

The aerosol optical properties $\sigma_{ep}$, SSA, AAE, and SAE are measured in-situ at low RH. The values of $\sigma_{ep}$ retrieved from

HSRL-2 and simulated by all four models are reported at ambient RH; the UM-UKCA model additionally reports dry $\sigma_{ep}$.

Those properties most critical to the underlying cloud albedo – the cloud fraction and cloud optical thickness – are evaluated by comparing 2D

- mean warm cloud fraction ($CF_{warm}$)

- geometric mean warm cloud optical thickness ($COT_{warm}$)

as retrieved from the polar-orbiting MODerate resolution Imaging spectroradiometer (MODIS; $CF_{warm}$ and $COT_{warm}$) and geostationary Spinning Enhanced Visible and InfraRed Imager (SEVIRI; $CF_{warm}$ only) to those modeled.

## 2.2 Comparison transects, altitude bins and statistics

Comparisons are made along several transects of gridboxes (**Figure 1** and Supplemental **Table S.1**). The locations of the

transects are dictated by frequent research flight paths, which varied across the three years of the project. A decided focus of the ORACLES field campaigns was to devote about half of the P-3 flight hours in each year to sampling along "Routine" flight tracks (Redemann et al., 2021). The explicit goal was to sample the transect across a set of randomly-distributed days throughout the field deployment in order to build a dataset representative of the deployment month.

During ORACLES 2016 the Routine flights followed a diagonal latitude/longitude transect (Diagonal transect, **Figure 1**, terminating near Namibia). With deployment based out of São Tomé in 2017 and 2018, the Routine flights were along a north-south oriented track centered on 5E (Meridional2, **Figure 1**). The Routine flight pattern usually consisted of a series of in-transit profiles and horizontal legs in the aerosol layer and in the boundary layer clouds. In 2017 and 2018, with the HSRL-2 lidar on board the P-3, the south-bound leg on Routine flights was usually flown at an aircraft maximum altitude

(approximately 5-6 km) to survey the aerosol and cloud layers below.  The north-bound run was then a combination of vertical profiling, horizontal legs, and sawtooth legs (for clouds). Each Routine flight included a different combination of legs and profiles, so only the latitude/longitude line of the flights (not their altitudes) were common to all. In 2016, on most Routine flight days the NASA ER-2 would also fly along the Routine track and, in some cases, overfly the P-3.

In addition to the dedicated Routine track in 2016, a significant number of P-3 Target of Opportunity flights (Redemann et al., 2021) were flown along a north-south transect near the southern African coast. As such, the Meridional1 (**Figure 1**) set of comparison gridboxes is also selected. Finally, for all three years a Zonal transect is established running from Ascension Island to the west African coast. The Zonal transect is located approximately at the latitudinal center of the southern African



biomass burning plume, along the northern edge of the main stratocumulus deck. Free troposphere transport in this region is driven by the southern African Easterly Jet, which is centered around 8°S (Adebiyi and Zuidema, 2016); as such, in the free troposphere the Zonal transect, to first order, covers a gradient in age from east (younger) to west (more aged).

The Zonal transect is located significantly farther from the deployment bases than the other comparison transects, so most gridboxes in this transect have little P-3 data. In 2016, only the ER-2 had sufficient sampling for meaningful comparisons with models along this transect. In 2017, when the P-3 did a "suitcase" flight to Ascension Island (Redemann et al., 2021) there was some coverage of the western-most Zonal gridboxes. For the Zonal transect, the only aerosol observations included in the comparison are profiles of $\sigma_{ep}$ from the HSRL-2 on board the ER-2 in 2016. In all three years, comparisons of $CF_{warm}$ and $COT_{warm}$ are included along the Zonal transect, since these observations are from satellite and so have good statistics in all three years.

In the discussions below, gridboxes are numbered from northwest to southeast for the Diagonal transect, north to south for the two Meridional transects, and west to east for the Zonal transect (**Figure 1**). Averages within each deployment year cover the following dates:

- August through 27 September, 2016;
- 9 August through 2 September, 2017; and
- 24 September through 25 October 2018.

These include the transit flights from Namibia (2016) and to and from São Tomé (2017 & 2018). For ease, we refer to these as the September 2016, August 2017 and October 2018 monthly averages.

Observed and modeled statistics are calculated for 500 m deep altitude bins from the surface to 6km, with two exceptions. Relative humidity is aggregated into 250m deep bins to more clearly show the transition from the boundary layer to the free troposphere. Light extinction from the HSRL-2 has 315 m vertical resolution, and this resolution is retained. Mean biases are calculated as average ratios in 1 km altitude bins for more robust statistics. For the in-situ observations, data are included in statistics whether made on level legs or during profiles, so the number of datapoints included in statistics can vary significantly with altitude within a given gridbox (**Figure 2**).

The aerosol properties compared here were measured from the aircraft, and so are available on specific days and for specific locations on each flight. In the aerosol comparisons, the model statistics used are calculated for only those dates and locations where the aircraft was present. In contrast, the observed $CF_{warm}$ and $COT_{warm}$ statistics are from satellite-based measurements (Section 3.2) and so are available for every day of each year's deployment. In this case, both observed and model statistics are calculated for every day of the deployment period across every comparison gridbox.

**Figure 2.** The number of minutes the P-3 aircraft spent in each gridbox and 500m altitude bin for the comparison transects shown in Figure 1. The 2016 Diagonal and 2017 and 2018 Meridional2 transects cover "Routine" flight tracks, which targeted semi-random sampling. Not all in-situ measurements have data available for all times, so the number of minutes of available data may be less than the number of minutes the P-3 was present in a given gridbox and altitude bin.



To test for the representativeness of the observed aerosol properties, for some variables two sets of modeled statistics are calculated for each gridbox and altitude layer: the first for only those locations and times when the aircraft was present, and the second for all daylight hours (defined here as 06:00-18:00 UTC) across the duration of that years' deployment (see Section 1.0 for date ranges). Comparison of the two allows for testing the representativeness of the observations for assessing monthly averages, assuming the model realistically captures aerosol variability. The number of minutes when the P-3 was present in each gridbox and 500m altitude bin is shown in **Figure 2**.

Shinozuka et al. (2020) also tested the representativeness of observed column properties (aerosol optical depth) for the September, 2016 campaign only. The vertically-resolved data provide additional, more detailed information on representativeness, something columnar passive satellite observations cannot provide. In addition, here representativeness is tested for all three campaign years.

## 3 Dataset descriptions

### 3.1 Observed aerosol properties, CO concentrations & relative humidity

Detailed descriptions of the instruments used to measure aerosols and gases are given in Appendix 9.1 of Shinozuka et al. (2020). Here, the characteristics of each measurement most relevant to the presented comparison are discussed. All in-situ observations are derived from the 1 sec resolution data collected on the P-3, available on the NASA public data archive (see Data Availability). Several of the measurements used here (e.g., absorption; see below) are very noisy at this resolution. To reduce noise the 1 sec resolution data are smoothed using a weighted average, calculated with a Gaussian weighting function covering ±30 sec on either side of each 1 sec resolution data point. The weighting function has 61 values, with the peak at value 31. The standard deviation was set to 12; this produces a weighting function such that the data points at time $t$-30 sec and $t$+30sec are weighted at 4.4% of the value at time $t$. Values of in-situ $\sigma_{ep}$, SSA, SAE and AAE are derived after this smoothing function is applied to the scattering and absorption data. In all cases, statistics for a given altitude bin and comparison gridbox are included for the in-situ observations only if at least 10 min of data in total are available.

Aerosol optical properties were measured in-situ at low (<40%) RH via an aerosol inlet with a 50% cut-off diameter of approximately 5μm (McNaughton et al., 2007). Above the boundary layer, the aerosol during ORACLES was dominated by accumulation mode biomass burning smoke, so it is expected that the in-situ instruments capture the properties of the vast majority of aerosol contributing to column radiative impacts and all of the biomass aerosol.

Carbon monoxide was measured with an ABB/Los Gatos Research CO/$CO_2$/$H_2O$ Analyzer modified for flight operations, with a precision of 0.5 ppbv for 10 sec averages (Liu et al., 2017; Provencal et al., 2005). Black carbon was measured as refractory BC (rBC) using a Single Particle Soot Photometer (SP2) (Schwarz et al., 2006; Stephens et al., 2003) calibrated





with Fullerene soot. The SP2 measurement of rBC mass is estimated to have an uncertainty of 25% at the provided 1sec resolution. A High-Resolution Time of Flight Aerodyne aerosol mass spectrometer (HR-ToF-AMS), operated in high sensitivity V-mode, was used to measure organic aerosol (OA) mass with an estimated accuracy of 50% at 1 second time resolution.


Aerosol light scattering ($\sigma_{sp}$) at 450, 550, and 700 nm was measured on board the P-3 at low (<40%) RH with a TSI model 3563 nephelometer, with the corrections of Anderson and Ogren (1998) applied. In 2018, two TSI nephelometers were operated, with one periodically measuring sub-micron aerosol only. When both were measuring the total aerosol, reported $\sigma_{sp}$ is the average of the two. In 2018, the 700 nm channel on the nephelometer wasn't working, so SAE data are not

available for that year.

As discussed below, most models report aerosol optical properties at ambient RH. Relative humidity profiles and aerosol hygroscopic growth factors inform whether this could be a significant source of differences between the modeled and observed aerosol optical properties and so are shown here. The observed ambient RH was calculated based on dew point

measured using an Edgetech 137 Vigilant hygrometer. Hygroscopic growth factors for 530nm light scattering were quantified during ORACLES using a pair of Radiance Research nephelometers, run at low (<40%) RH and approximately 85% RH respectively. However, there were instrumental issues that resulted in significant data gaps in 2016 and 2018, and instrumental problems across the full 2017 campaign. This complicates correcting to humidified scattering values for the statistical comparisons with the models. As such, here we use these data only to estimate the effect of humidification on

scattering, based on aerosol characteristics aggregated across all observations (not just those in the comparison transects) within each field season.

Dry aerosol light absorption ($\sigma_{ap}$) at 470, 530, and 660nm was calculated using measurements from one (2017 and 2018) or two (2016) three-wavelength Radiance Research Particle Soot Absorption Photometers (PSAPs). For 2016, the values from

the two PSAPs are averaged; for 2017 and 2018 only one of the PSAPS consistently measured the total ambient aerosol absorption, so only data from that PSAP was used. Filter-based absorption measurements such as the PSAP are known to have loading-based artefacts that produce a positive bias that requires correction (e.g. Bond et al., 1999; Virkkula, 2010). The original PSAP instrument measured $\sigma_{ap}$ at only one wavelength, 530nm, so correction factors at this wavelength are better understood than at 470 and 660nm, where they are untested for accuracy. Here, two sets of correction factors have

been applied to the PSAP data: the wavelength-averaged and the wavelength-specific corrections, both described in Virkkula (2010). These correction factors are very similar at 530nm but yield different values of $\sigma_{ap}$ at 470nm and 660nm. They therefore yield different values of derived absorption Ångström exponent, but nearly identical 530nm SSA.





Scattering at the 450, 550, and 700 nm wavelengths ($\lambda$) is used to calculate a linear fit to $\log(\sigma_{sp})$ versus $\log(\lambda)$, yielding the
scattering Ångström exponent, SAE. The absorption Ångström exponent, AAE, is analogously calculated from $\sigma_{ap}$ at 470nm, 530nm, and 660nm for, as noted above, $\sigma_{ap}$ derived using the two sets of Virkkula (2010) correction factors. The observed values of $\sigma_{ep}$ and SSA included here are at 530nm, for low-RH aerosol. They are calculated by adjusting the measured, low-RH 550nm $\sigma_{sp}$ with the above-calculated SAE. This adjusted $\sigma_{sp}$ is then summed with the 530nm $\sigma_{ap}$ to get $\sigma_{ep}$, and SSA is calculated as the ratio of 530nm $\sigma_{sp}$ to 530nm $\sigma_{ep}$. SAE and SSA are calculated only when $\sigma_{ep}$ is greater than 10 Mm$^{-1}$ and 295 AAE is only calculated when $\sigma_{ap}$ is greater than 5 Mm$^{-1}$ in order to avoid including data dominated by noise.

All of the above measurements were made from the P-3 aircraft. Data from the airborne second-generation High Spectral Resolution Lidar version 2 (HSRL-2) that was flown on the ER-2 aircraft in 2016 and the P-3 in 2017 and 2018 are also included in the comparison. The HSRL-2 is a remote-sensing instrument so retrieved values of $\sigma_{ep}$ are at ambient RH, and 300 therefore are more directly comparable to the modeled values. The HSRL-2 independently detects backscatter from aerosols and molecules using the spectral distribution of the returned signal, thereby retrieving $\sigma_{ep}$ without having to make assumptions about the backscatter-to-extinction ratio of the aerosol (Shipley et al., 1983; Hair et al., 2008; Burton et al., 2018). The HSRL-2 retrieves $\sigma_{ep}$ at 355nm and 532nm with 315m vertical resolution; here we use the 532nm data only for comparison to modeled 550nm $\sigma_{ep}$.

## 305 3.2 Observed cloud properties

From the standpoint of aerosol forcing, the clouds of most interest in the SE Atlantic are Stratocumulus and Cumulus (warm, low) clouds in the boundary layer, as these clouds are most prevalent in the region and underlie the aerosol plume, so they are a strong controlling factor on the direct aerosol radiative effect sign and magnitude. As such, here we compare the warm, low cloud (<2.5km, T>273K) fraction (CF$_{warm}$) from models to that retrieved in several satellite products.

Cloud optical thickness for these clouds (COT$_{warm}$) is approximately lognormally distributed (e.g. Supplemental **Figure S.1**), so for COT$_{warm}$ we compare the geometric mean of all values within the comparison gridboxes. This statistic was selected as the most physically meaningful, since it more closely represents the cloud optical thickness, and therefore cloud impact on scene albedo, for heterogeneous scenes.

## 315 3.2.1 MODIS-Standard cloud products

The Collection 6 MODIS Level 3 (L3) daily cloud products (Platnick et al., 2015a) from both the Aqua (MYD08) and Terra (MOD08) satellites are used to calculate average warm cloud fractions, CFwarm. These L3 products are statistical aggregations at 1˚x1˚ resolution (latitude  x longitude) of the MODIS Level-2 pixel-level cloud retrievals (Platnick et al., 2015b; Platnick et al., 2017). Since Aqua and Terra are polar-orbiting satellites, their cloud retrieval statistics from the





ORACLES comparison gridboxes are from, on average, 10:20 UTC (Terra) and 13:40 UTC (Aqua). Herein we refer to these as the "MODIS-Standard" retrieved cloud properties.

The L3 MODIS variables used for CFwarm are Cloud_Retrieval_Fraction_Liquid and Cloud_Retrieval_Fraction_PCL_Liquid, with the latter allowing for inclusion of partly-cloudy pixels. Data are excluded

from statistics if the retrieved cloud top height is greater than 2.5 km, in order to include only low warm clouds. These variables only include the Level-2 pixel population that is identified as liquid phase, overcast, and that has successful cloud optical property retrievals, allowing classification as liquid clouds. As such, CFwarm may be smaller than the actual warm cloud fraction, depending on the rate of cloud optical property retrieval failure (see, e.g., Cho et al., 2015) and the prevalence of broken clouds and cloud edges in a retrieval pixel.  However, for the selected comparison transects – and for this region in

general – the fraction of mid-level and high clouds is low. For example, in 2016 warm, low clouds comprise, on average, 93% or more of the clouds in the Diagonal and Meridional1 transect gridboxes and the four eastern-most Zonal transect gridboxes. In the seven western-most Zonal gridboxes >99% of the clouds are warm clouds. An exception is the gridboxes closer to the African coastline, where mid-level clouds in particular can be more frequent. This is consistent with the fact that most mid-level and high clouds in the region originate over the continent (Adebiyi et al., 2020), a phenomenon we observed

directly during the field campaigns.

### 3.2.2 MODIS-ACAERO cloud products

Retrievals of cloud properties from satellite imager-based observations can be affected by the presence of aerosol above the clouds, particularly when that aerosol is light-absorbing (Haywood et al., 2004; Coddington et al., 2010; Meyer et al., 2013). While retrieved CF is not significantly impacted, the retrieved COT will be. If not accounted for, the attenuation of cloud-

reflected solar radiation due to aerosol absorption can be interpreted by satellite imager cloud retrieval algorithms as higher effective radii and as a lower COT. Therefore, in addition to the MODIS-Standard cloud retrievals we calculate cloud statistics using the L2 (1 km resolution) MOD06/MYD06ACAERO retrievals from MODIS that use the Meyer et al. (2015) approach, which accounts for the effects of the absorbing aerosol layer above low clouds. These retrievals, referred to here as MODIS-ACAERO, simultaneously retrieve the above-cloud aerosol optical properties and the unbiased cloud optical

properties, and are used as the reference for observed COTwarm. (Specifically, the Cloud_Optical_Thickness_ModAbsAero parameter is used). The MODIS-ACAERO retrieved CFwarm is also included, which differs from the MODIS-Standard definition in its use of cloud-top height (CTH) as an additional filter (specifically, CTH < 4km, thus excluding mid-level clouds). Otherwise, as with the MODIS-Standard retrievals, these are averages from the MODIS instruments on the Terra and Aqua satellites.



### 3.2.3 SEVIRI-LaRC cloud products


Warm clouds over the SE Atlantic have a significant diurnal cycle, particularly in cloud fraction (Rozendaal et al, 1995; Wood et al., 2002; Painemal et al., 2015). A question is whether the MODIS retrievals, which make observations only twice daily, are representative of the daytime averages. The Spinning Enhanced Visible and Infrared Imager (SEVIRI) on the geostationary satellite Meteosat-10 views the SE Atlantic region at all times of day. We use the cloud fraction retrieved from

SEVIRI for three purposes: First, we calculate the average daytime $CF_{warm}$ in each comparison gridbox, for testing the modeled average daytime $CF_{warm}$. Second, we calculate the difference in the average daytime $CF_{warm}$ and the average of $CF_{warm}$ at 10:30UTC and 13:30UTC only, as an estimate for how different $CF_{warm}$ from the MODIS Terra and Aqua retrievals might be from an actually full daytime average of $CF_{warm}$. Third, as described below, we use the diurnal cycle in $CF_{warm}$ to infer the diurnal cycle in $COT_{warm}$, and therefore the representativeness of the MODIS-ACAERO Terra+Aqua

$COT_{warm}$ to the daytime average.

Here, the SEVIRI retrievals described by Minnis et al. (2008; 2011a; 2011b) and Painemal et al. (2015) are used and are referred to as the SEVIRI-NASA Langley Research Center (LaRC) retrievals. Warm cloud fractions are derived at 0.25° grid resolution from pixel-level (3 km) retrievals by counting pixels with "liquid" cloud phase and the effective cloud top

temperature $T_{cldtop}>273.2K$. Retrievals are provided every 30 minutes. We limit our analysis to daytime samples with solar zenith angles (SZA) of less than 75˚ to minimize retrieval uncertainties in the day-night transition.

In this region, cloud cover tends to be at a maximum in early morning, then either decreases throughout the day or decreases until mid- to late afternoon and then again increases (**Figure S.2**; Painemal et al., 2015). The average of $CF_{warm}$ at

10:30+13:30 is generally lower than but within 5% of the daytime average (**Table S.2**). The exception is at the north end of the Meridional1 and Meridional2 transects, when the 10:30+13:30 average is up to 14% below the daytime average. While the 10:30+13:30 average $CF_{warm}$ is lower than the daytime average, it does represent well $CF_{warm}$ mid-day, when solar flux (and therefore radiative forcing) is at a maximum.

An additional question is whether the $COT_{warm}$ values from the 10:30+13:30 MODIS-ACAERO retrievals are representative of the daytime average. The SEVIRI-LaRC retrievals don't simultaneously provide aerosol optical depth and cloud products, and inferred $COT_{warm}$ could be biased if there is a high-AOD aerosol layer above the clouds. To approximate the diurnal cycle in $COT_{warm}$ an empirical fit to $COT_{warm}$ versus $CF_{warm}$ from the MODIS-ACAERO dataset from all three field campaign years and comparison transects was used to approximate the difference between the

10:30+13:30 average $COT_{warm}$ and the average of $COT_{warm}$ across the full daytime. The resulting fit (**Figure S.3**) is

$$COT_{warm,fit} = 1.663 \times e^{1.982*CF_{warm}} \tag{1}$$





COT$_{warm}$, like CF$_{warm}$, is slightly lower –typically by less than 0.5 – for the 10:30+13:30 average than for the full daytime average, when calculated using the approximation in Eqn. [1] (**Table S.3**). At the northern end of the 2016 Meridional1 and 2017 Meridional2 transects the difference is closer to 1.0. For a solar zenith angle of 30°, a decrease in COT$_{warm}$ from 10.0, which is typical of clouds in this region (see Section 4.4), to 9.0 reduces cloud albedo by only 0.02, from 0.46 to 0.44 for (see Section 6.0). The influence on scene albedo, which is the variable of interest for DARE, will be even smaller any time CF$_{warm}$ is less than 1.0.

### 3.3 Modeled aerosol and cloud fields

Data for all three ORACLES years are available for the WRF-CAM5 and GEOS models; UM-UKCA and ALADIN provided comparison data for the 2016 and 2017 ORACLES field campaign periods only. Statistic for all variables listed in Section 2.1 are provided for the WRF-CAM5 model. Statistics are not provided for RH from GEOS; for CO from UM-UKCA; and for CO, RH and AAE for ALADIN.

All models report aerosol optical properties at ambient RH, in contrast to the observed optical properties which are at low RH. The UM-UKCA model also reports dry aerosol optical properties. In addition, all models report extinction and SSA at 550nm, whereas the observed values are at 530nm. Finally, the modeled AAE and SAE are calculated from 400 to 600nm, whereas the observed AAE is calculated using $\sigma_{ap}$ at the three wavelengths of 470, 530 and 660nm, and the SAE using $\sigma_{sp}$ at 450, 550 and 700nm.

The reported model CF$_{warm}$ values are the mean of the gridbox 2D warm, low cloud fractions, i.e. the fraction of the gridbox covered by cloud as viewed from above, not the fraction of the 3D gridbox filled by cloud. Modeled CF$_{warm}$ values exclude mid- and high-altitude clouds, and includes all low-lying warm clouds. This 2D CF is roughly equivalent to what would be observed via satellite and is the relevant quantity when interested in short-wave radiative forcing. For all models, 3-D cloud fractions are converted to 2-D warm, low cloud fractions (CF$_{warm}$) by assuming maximum horizontal overlap in clouds at different altitudes within the same model column. As with the observed mean COT$_{warm}$ values, the model mean COT$_{warm}$ for each gridbox is the geometric-mean (Section 3.2) with one exception: for the ALADIN model, the geometric mean COT$_{warm}$ statistic was not calculated, so the median COT$_{warm}$ is used instead.

Details on each model are given in Section 9.2 of Shinozuka et al. (2020), with brief descriptions given here. WRF-CAM5 is the regional Weather Research & Forecasting model with chemistry (WRF-Chem) coupled with the Community Atmosphere Model v.5 (CAM5) physics (Ma et al., 2014) with updated aerosol activation parameterizations (Zhang et al., 2015a,b). Here, the model is run at 36 km horizontal resolution and with 74 vertical layers varying in resolution from 10 m to 500 m, with higher resolution at lower altitudes. Aerosol mass and number are tracked, and aerosol optical properties calculated with Mie code assuming an internally mixed aerosol with three aerosol modes (Aitken, accumulation and coarse). Cloud





formation is driven by the shallow convection scheme of Bretherton and Park (2009) and deep convection by the Zhang and McFarlane (1995) scheme, with interactive aerosols. Smoke emissions are initialized daily from the Quick Fire Emissions Data set version 2 (QFED2) (Darmenov and Da Silva, 2015) which provides emissions on a daily basis. Smoke is emitted directly into the boundary layer without using any plume injection parameterization. The model is initialized every five days using the NCEP Final Operational Global Analysis (FNL) and CAMS reanalysis and runs for 7 days, with the first two days

of the run used for spin-up. Data are output at three-hour time resolution and aggregated for statistics.

The GEOS (Goddard Earth Observing System v. 5) global model (Molod et al., 2015; Rienecker et al., 2008), often referenced as GEOS-FP (GEOS Forward Processing), is the forecast system of NASA's Global Modeling and Assimilation Office. It is run in near real time at approx. 25 km horizontal resolution (0.25° in latitude, 0.3125° in longitude) and 72

vertical layers (of which 25 layers are between the surface and 400 hPa). The model is initialized every 12 hours, with aerosol fields saved every three hours and cloud fields hourly. The model is initialized using the MERRA-2 reanalysis product, so it includes assimilation of observed aerosol optical depth (AOD) data. This 'nudging' towards observed AOD should improve this model's simulated $\sigma_{ep}$ values relative to a free-running model. Like WRF-CAM5, GEOS also uses QFED2 biomass burning emissions and injects the emissions at the surface. It prognostically predicts CO, aerosol

component masses and ambient-RH aerosol optical properties using GOCART (Goddard Chemistry Aerosol Radiation and Transport; Chin et al., 2002; Colarco et al., 2010). It assumes that aerosols are externally mixed in modes of fixed mean diameter and standard deviation. Their optical properties are computed for each aerosol species included in GOCART and as a function of RH (Randles et al., 2017; Colarco et al., 2014). GEOS assimilates AOD observations from remote sensing every three hours (Albayrak et al., 2013). Organic aerosol (OA) concentration is not provided explicitly by GEOS, but

organic carbon (OC) is. The ratio OA/OC=1.4 is used to obtain the reported OA concentrations. Both hydrophilic and hydrophobic BC and OC are simulated; masses reported here are from the sum of the hydrophilic and hydrophobic components. Clouds are simulated by the convective parameterization.

The Unified Model coupled to the United Kingdom Chemistry and Aerosol model (UM-UKCA) is a global model that

forecasts aerosols and clouds and is run here with a configuration modified from that used in Gordon et al. (2018), which also focused on the SE Atlantic. The model resolution varies with latitude (N216 resolution), with approximately 60 x 90 km resolution at the equator. It has 70 vertical levels between the surface and 80 km altitude, with a decreasing vertical resolution such that the grid spacing at 1.5 km altitude is approximately 200 m. It is nudged to horizontal wind fields (not to temperature) from ERA-interim reanalyses, with nudging starting at 1700 m above the surface and ramping up to its full

strength at 2150 m altitude. The reanalysis files are read in every six hours, which is also the relaxation time for the nudging. The model is run continuously forward from the initialization used by Gordon et al. (2018). In contrast to WRF-CAM5 and GEOS, biomass burning emissions are updated daily using the FEER inventory (Ichoku and Ellison, 2014). Smoke aerosol is emitted into and distributed through the boundary layer, such that concentrations are highest at the surface, then taper down





to zero at 3 km above the surface (Gordon et al., 2018). The emitted smoke has an initial log-normal size distribution with a
mode centered on 120 nm diameter. Sea-salt emissions are based on winds, no dust emissions are included, and all other
emissions are from the CMIP5 inventories. Aerosols in the model are represented in five size modes of internally-mixed
aerosol. Both dry and ambient-RH aerosol properties are tracked, with hygroscopicity based on Petters and Kreidenweis
(2007). Convection is represented using the pc2 sub-grid cloud scheme of Wilson et al. (2008) or is parametrized where it
cannot be resolved.


The ALADIN model is a regional climate model developed at Météo-France/CNRM. The v.6 version used here has a more
detailed treatment specifically of biomass burning aerosols than previous versions (Mallet et al., 2019, 2020). The model has
12 km horizontal resolution and 91 vertical levels with 28 located between the surface and 6 km altitude. Lateral boundary
conditions and the initial state for the modeled region come from the ERA-Interim reanalysis (Dee et al. 2011). The model
includes TACTIC (Tropospheric Aerosol Scheme for Climate In CNRM; Nabat et al. 2020) which includes sea salt, desert
dust, sulfates, black and organic carbon separated in 12 aerosol size bins. All emissions come from the CMIP6 emissions
inventory (van Marle et al. 2017), which uses the GFED database for biomass burning emissions. This inventory has realistic
biomass burning emissions only through 2014, so these runs were done using constant year 2014 emissions. Further, while
the BC emissions from GFED are used, ALADIN uses a fixed particulate organic matter (POM) to organic carbon (OC) ratio
based on Formenti et al. (2003), so secondary organic aerosol formation is not accounted for. The radiative properties of
liquid clouds are calculated in the shortwave using the Slingo and Schrecker (1982) parametrizations. The atmospheric
physics has recently been revisited, as described in detail in Roehrig et al. (2020). For the model runs used here, the first
indirect effect was not simulated, and cloud droplet effective radius was held fixed at 10μm.

## 4 Results

The representativeness of the sampled aerosol properties to that of the entire field campaign period within each deployment
year are discussed in Section 4.1. Biases in modeled extensive properties (CO, BC, and OA concentrations and $\sigma_{ep}$) are then
discussed in Section 4.2, in aerosol intensive properties (SSA, SAE, and AAE) in Section 4.3, and in clouds (CF$_{warm}$ and
COT$_{warm}$) in Section 4.4.

### 4.1 Representativeness of observations

As discussed above, a goal of the Routine track sampling was representative sampling rather than, e.g., targeting high-
concentrations plumes. With limited flight hours, and in-situ sampling from the P-3 at specific altitudes on each flight track,
the number of minutes spent in many of the gridboxes and altitude bins was on the order of 1-2 hours in total over the
approximately month-long campaign in each year; for some gridboxes and altitudes it was <20 min (**Figure 2**). The amount
of data collected is particularly limited at the far reach of the comparison transects, i.e. the northwestern-most and northern-





most gridboxes in the 2016 Diagonal and Meridional1 transects, and the southern-most gridboxes in the 2017 and 2018

Meridonal2 transect. For the Zonal transect, in-situ sampling was extremely limited, with significant sampling only in 2017

in the western-most Zonal gridboxes 1 and 2 (from the "suitcase" flights to Ascension Island; Redemann et al., 2020) and in

gridbox 11, which intersects with the Meridional2 Routine track.

**Figure 3** shows the ratio of the average of $\sigma_{ep}$ in the model for those times when the aircraft was present for sampling (in-

situ for the P-3 and of the full column below the aircraft for the HSRL-2 on the ER-2 aircraft; i.e. "sampled") to the daytime

average for the full duration of the field deployment that year ("climatology"). Shinozuka et al. (2020) tested the

representativeness of the observed column properties to the full month of the 2016 campaign period using as a metric the

mean bias (MB) and the root mean squared deviation (RMSD) of CO and aerosol properties, along with their ratio (%) to the

monthly mean. They calculated MB and RMSD across gridboxes for data within broad altitude ranges, including for 3-6km.

Here we test for representativeness through ratio of the means in 1km deep altitude bins for each gridbox (colored dots in

**Figure 3**). This metric is the same as MB(%)/100+1. This selection was made because the RMSD gives greater weight to

individual large deviations, and the focus of this paper is on the average bias in observed values, which will most directly

scale with a mean bias in DARE. In addition to calculating the mean bias for each gridbox, the transect-mean bias is

calculated across all gridboxes in a given comparison transect and altitude bin (open circles in **Figure 3**).

Both WRF-CAM5 and GEOS indicate that $\sigma_{ep}$ at plume altitudes (2-5 km) along the 2016 Diagonal transect is, on average,

somewhat higher during the times sampled by the P-3 than it is for the monthly average (**Figure 3a,b**).  The transect-mean

difference is up to 50% depending on the altitude and model, with WRF-CAM5 showing larger and less variable differences

than GEOS. Values of $\sigma_{ep}$ at the times measured by the HSRL-2 from the ER-2 better represented the month-long average

(**Figure 3c,d**), with mostly moderate (<20%) differences in the two according to GEOS. In WRF-CAM5, the ratio of

sampled $\sigma_{ep}$ to the monthly-long "climatology" increases with altitude from 2 km to 6 km, indicating that the sampled plume

may have been centered at higher altitudes than was typical for that month.

The 2016 Meridional1 transect is not a Routine flight track, so sampling was on flights targeting the smoke plume and/or

specific cloud fields and includes fewer observations than the Diagonal transect (**Figure 2**). As such, it was not expected to

be as representative of the monthly average. Despite this, the transect-mean difference in $\sigma_{ep}$ between the times measured by

the P-3 and the month-long average is <20% according to both models (**Figure 3e,f**). Both models do, however, indicate that

smoke concentrations sampled by the P-3 at higher altitudes (4-6km) are much higher than was typical. Values of $\sigma_{ep}$ from

times when the HSRL-2 could make observations from the ER-2 are more consistently representative of the month-long

average, with transect-mean differences in most altitude bins above 2 km <10-20% and <50% for almost all individual

gridboxes and altitudes. As for the 2016 Diagonal transect, the WRF-CAM5 simulations indicate the sampled plumes were

centered at a higher altitude than is typical for this month.
















**Figure 3.** The ratio in mean of $\sigma_{ep}$ modeled by WRF-CAM5 (left column) and GEOS (right column) for only those times when observed values are available ("sampled") to the mean of $\sigma_{ep}$ across the entire field campaign time period ("climatology") in the (a-d) Diagonal transect in Sept. 2016, (e-h) Meridional1 transect in Sept. 2016, (i-l) Meridional2 transect in Aug. 2017, (m-p) the Meridional2 transect in Oct. 2018, and Zonal transect in 2016 (q,r) and 2017 (s,t,u,v). The representativeness of both the in-situ and HSRL-2 observations of $\sigma_{ep}$ are shown. The color dots show the means within individual comparison gridboxes; open circles are the mean across all gridboxes in that transect. Note that in panel f) there is a single gridbox datapoint that is off-scale (sampled:climatology $\sigma_{ep}$ >4).

The two models give very different results regarding the representativeness of the in-situ sampling by the P-3 along the Meridional2 transect in both 2017 and 2018. The WRF-CAM5 model indicates that $\sigma_{ep}$ in the 2-6km altitude range when the P-3 was present for sampling were generally within 20%, and almost always within 50% of the month-long climatology. GEOS simulations, however, show significantly higher values of $\sigma_{ep}$ in the P-3 sampling average than in the month-long climatology for almost all gridboxes and altitudes. This highlights the difficulty of testing for sampling representativeness based solely on modeled aerosol fields, which themselves may contain variability across a range of scales that does not necessarily match that of reality. Future studies may want to use a synthesis of modeled and satellite-retrieved properties (e.g. of AOD) for a more robust analysis of sampling representativeness.

The model average of $\sigma_{ep}$ from times/locations when HSRL-2 could make measurements along the Meridional2 transect in both 2017 and 2018 is, on average within 20% of the month-long climatology for the 2-5km altitude range in the WRF-CAM5 model. The GEOS model simulated a much larger difference between the average for HSRL-2 "sampled" times and the month-long "climatology" along the Meridional2 transect, particularly in 2018. It also shows much greater variability across the different gridboxes in the sampling bias, again likely highlighting differences in simulated aerosol variability within the different models.

Typically, the HSRL-2 retrievals are available in full "curtains" from just below the aircraft flight level to either the surface or cloud top from the south-bound. In 2016, the HSRL-2 was on the ER-2, which always flew fully above the plume, so it captured the full vertical extent of the plume. In 2017 and 2018, when it was on board the P-3, the HSRL-2 generally could capture most of the plume vertical extent during the out-bound leg of Routine flights along the Meridional2 transect, since they were flown at high altitude. A combination of in-situ measurements and HSRL-2 measurements would then be collected on the return, north-bound leg, which was flown at a variety of altitudes. As such, there are more data from the HSRL-2 than from the in-situ measurements to contribute to comparison statistics. This likely explains why $\sigma_{ep}$ from the times the HSRL-2 could make measurements is generally more representative of the month-long average than $\sigma_{ep}$ from the times in-situ observations are available (**Fig. 3k,l** vs **Fig. 3i,j** and **Fig. 3o,p** vs **Fig. 3m,n**).



In 2016, the ER-2 flew along the Zonal transect on several flights (**Figure 1**). WRF-CAM5 and GEOS both simulate average $\sigma_{ep}$ from HSRL-2 sampling times that are within 20% of the month-long average in the 2-5 km altitude range along the 2016

Zonal transect (**Figure 3q,r**). The modeled "sampled" to "climatology" ratio of $\sigma_{ep}$ is both more positive and more variable across gridboxes for 4-5km than for 2-4km or 5-6km. This likely reflects the sampling coincidence with individual elevated plumes during the ER-2 flights.

In 2017, $\sigma_{ep}$ from both the in-situ and HSRL-2 sampling times poorly represent the August average for most gridboxes and

altitudes (**Figure S.4**), and in 2018 the P-3 did not fly along the Zonal transect. For this reason, comparisons are not made of modeled and measured aerosols along the Zonal transect. Clouds (CF_warm and COT_warm) were measured by satellite on all field campaign days, so comparisons of these fields along the Zonal transect are included for all three years.

Tests of the representativeness of $\sigma_{ep}$ measured from the aircraft addresses sampling biases in the concentration of the

aerosol. In the context of DARE calculations, an additional question is whether the optical properties of the sampled aerosol are representative. Aerosol SSA in particular is a strong controlling factor for the sign and magnitude of DARE. In the WRF-CAM5 model SSA of the aerosol in the 2-6 km altitude range at the times when there are in-situ measurements from the P-3 are generally within 0.01 of the month-long average for that campaign year (not shown). SSA deviations from the average were a bit larger in the GEOS model in some gridboxes at these altitudes. In particular, in the Meridional2 transect the

aerosol measured in the two southern-most gridboxes in 2017 has an anomalously low SSA (by about 0.03-0.04) and in 2018 the SSA for 4-5km is similarly anomalously high. These two gridboxes were the most under-sampled in the Meridional2 transect, since they were the farthest from the deployment base. As will be seen below the observed SSA varied more than the WRF-CAM5 modeled SSA, but less than the GOES modeled SSA, so the apparent representativeness of the sampled aerosol SSA may be a reflection of an inherent invariance in SSA in the models rather than an indication of the actual

representativeness of the sampled aerosol optical properties.

**4.2 Biases in plume extensive properties**

Biomass burning smoke from the African continent is advected over the SE Atlantic largely in the free troposphere, and this is reflected in the observed profiles of $\sigma_{ep}$ across the comparison transects (**Figures 4** and **5**). This continental airmass carries with it water vapor (Pistone et al., 2021), though RH in the plume is still generally less than 60% in September 2016 and

August 2017 and less than 70% in October 2018, except in the two northern-most gridboxes of the Meridional2 transect in 2018 (**Figure 6**). The impact of humidification on $\sigma_{ep}$ and on this comparison is discussed in Section 4.2.3.








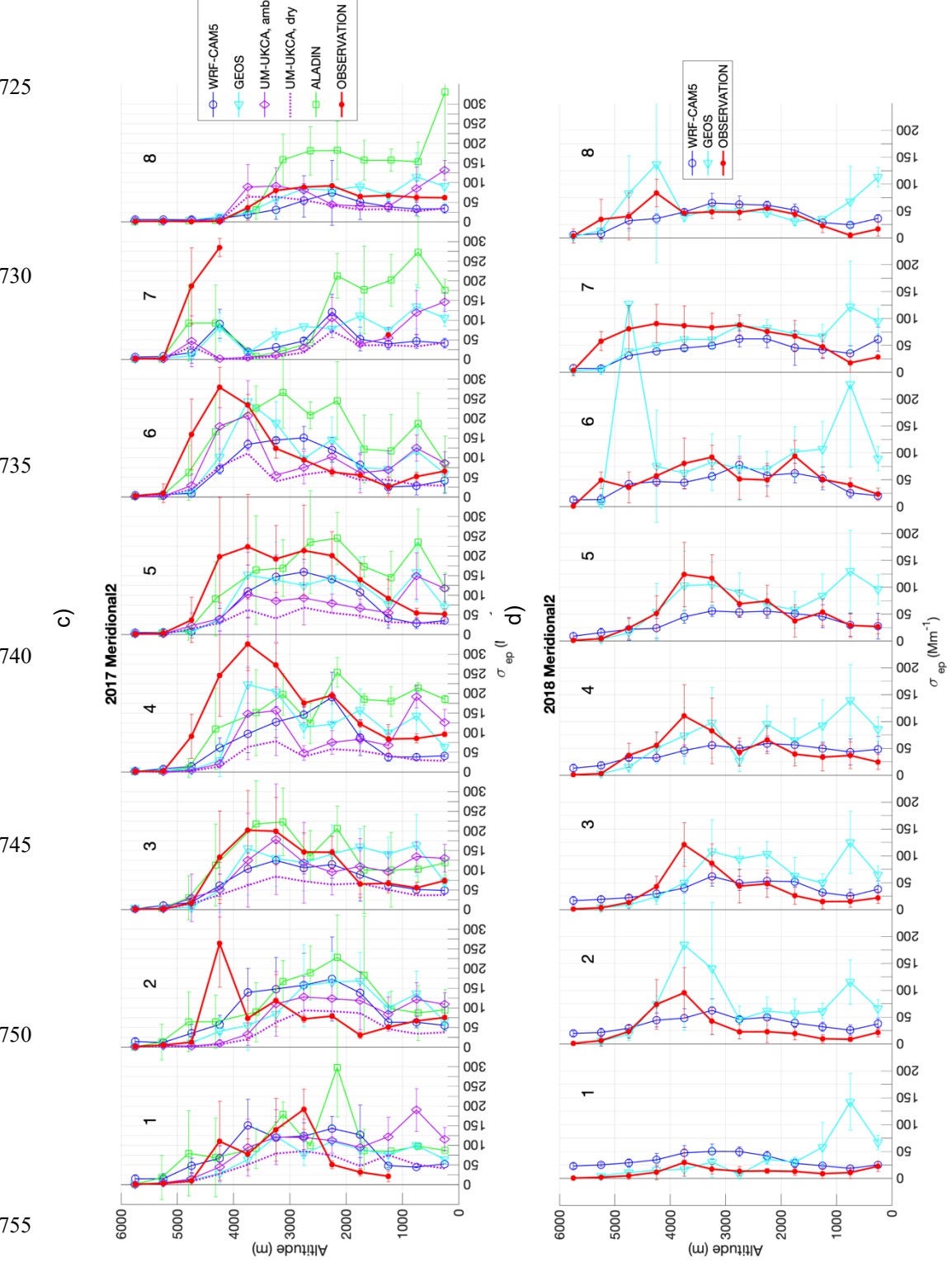

**Figure 4.** Profiles of $\sigma_{ep}$ as observed in-situ from the P-3 aircraft (dry, 530nm) and modeled (ambient-RH, 550nm) for the a) Diagonal transect, Sept. 2016, b) Meridional1 transect, Sept. 2016, c) Meridional2 transect, Aug. 2017 and d) Meridional2 transect, Oct. 2018. Shown are means and standard deviations calculated across only those times and locations when in-situ measurements were made. Gridboxes within these and subsequent plots are ordered left to right from north/west to south/east and are numbered as in Figure 1.









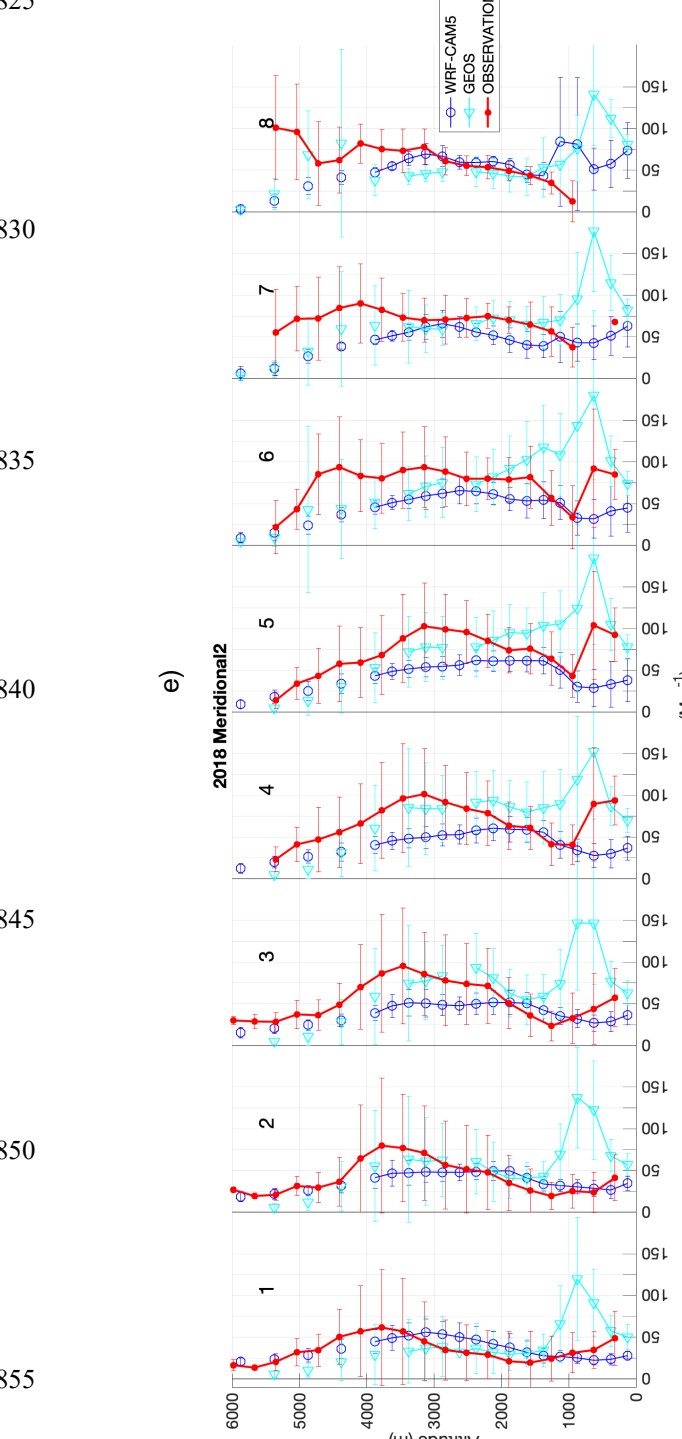

**Figure 5.** Profiles of $\sigma_{ep}$ retrieved from HSRL-2 lidar data from the ER-2 aircraft (532nm) and modeled (550nm) for the a) Diagonal b) Meridional1 and c) Zonal transects in Sept. 2016; and for the Meridional2 transect in d) Aug. 2017 and e) Oct. 2018. Statistics are calculated across only those times and locations when the ER-2 was present, though observations are not always available (e.g. 2016 Diagonal gridbox 8, which is partly over land). HSRL-2 data are shown at the reported altitude resolution of 315m.









**Figure 6.** As in Figure 4, but for RH, and at 250m vertical resolution rather than 500m vertical resolution.



Analogous figures showing profiles of the other extensive variables can be found in Supplemental **Figures S.5** (CO), **S.6** (BC), and **S.7** (OA). In the sections below, the modeled-to-observed ratios of these parameters are discussed; these should be viewed in the context of the smoke plume distribution (Figures 4 and 5) since large biases in the core of the smoke plume have much greater impact than large biases where concentrations are low.

### 4.2.1 Carbon Monoxide (CO)

Carbon monoxide does not lead to climate forcing, but it is an excellent and relatively inert tracer of biomass burning emissions and so is discussed here. WRF-CAM5 modeled CO at plume altitudes (2-5 km) is typically around 70% to 80% of that observed, with a slightly greater low bias in 2018 (**Table 1** and **Figure S.5**). GEOS also has a low bias in CO at plume altitudes, but the biases are somewhat smaller and were more variable than for WRF-CAM5. In 2016 and 2017 the GEOS CO concentrations are increasingly biased low going from 2 km to 5 km altitude. In 2018, the

GEOS biases are more consistent at 60-80% of the observed CO concentrations across almost all altitudes and gridboxes. The GEOS modeled plume extends to lower altitudes than observed (**Figure S.5**), so that for the 1-2 km altitude bin the overall low bias in modeled CO is effectively offset by the contribution of the lower part of the modeled plume. Near the surface (0-1 km), CO in both WRF-CAM5 and GEOS is biased low. CO was not reported for the UM-UKCA and ALADIN models. The biases in WRF-CAM5 and GEOS suggest under- estimates in CO

emissions or possibly in the efficiency of transport of the biomass burning plume over the SE Atlantic from the burning source regions, since CO is not affected by scavenging processes. An earlier evaluation by Das et al. (2017) of GEOS simulations of the SE Atlantic biomass burning plume compared to CALIOP lidar profiles indicated that, for that model, transport biases are the more likely explanation.

### 4.2.2 Aerosol BC and OA masses

During ORACLES, the aerosol components BC and OA were measured in-situ and were reported for the WRF-CAM5, GEOS and UM-UKCA models. In addition to emissions and transport processes, accurate simulation of aerosol concentrations requires simulating loss processes, including dry and wet deposition and any in-atmosphere production or loss. During the biomass burning season (July-October), south of ~2-3S latitude in the SE Atlantic there are few clouds with tops above 2km, with small drop sizes further discouraging the wet scavenging of aerosols from the free troposphere

(Adebiyi et al., 2020). For the 2016 Diagonal comparison gridboxes, and for all but the northern-most two to three Meridional2 gridboxes in 2017 and 2018 almost all wet scavenging occurs in moist convection over the central African continent (Ryoo et al., 2021). Once over the ocean, wet deposition likely plays essentially no role in driving aerosol gradients in latitude and longitude above the marine boundary layer across our comparison transects, except possibly in Meridional2 gridboxes 1-3 (located between 0.5N and 5.5S). Fall speeds for accumulation mode aerosols are at most a few



**Table 1.** Median across each comparison transect of modeled-to-observed ratios for extensive parameters averaged within 1km altitude bins. Shown are median biases for CO concentrations, BC mass, OA mass and $\sigma$ep as measured in-situ from the P-3 aircraft. Modeled values of $\sigma$ep are at 550nm and ambient RH, whereas observed values are at 530nm and for dry aerosol.

| | WRF-CAM5 | | | | GEOS | | | | UM-UKCA | | | ALADIN | | |
|---|---|---|---|---|---|---|---|---|---|---|---|---|---|---|
| | 2016 Diag | 2016 Mer1 | 2017 Mer2 | 2018 Mer2 | 2016 Diag | 2016 Mer1 | 2017 Mer2 | 2018 Mer2 | 2016 Diag | 2016 Mer1 | 2017 Mer2 | 2016 Diag | 2016 Mer1 | 2017 Mer2 |
| **Carbon Monoxide (CO)** | | | | | | | | | | | | | | |
| **5-6km** | 0.60 | 0.80 | 0.72 | 0.74 | 0.76 | 0.80 | 1.00 | 0.82 | | | | | | |
| **4-5km** | 0.70 | 0.82 | 0.60 | 0.69 | 0.65 | 0.87 | 0.73 | 0.76 | | | | | | |
| **3-4km** | 0.74 | 0.72 | 0.87 | 0.65 | 0.72 | 0.91 | 0.89 | 0.66 | | | | | | |
| **2-3km** | 0.81 | 0.72 | 0.72 | 0.68 | 1.05 | 0.90 | 0.89 | 0.77 | | | | | | |
| **1-2km** | 0.76 | 0.68 | 0.68 | 0.68 | 1.01 | 1.02 | 1.12 | 0.77 | | | | | | |
| **0-1km** | 0.83 | 0.70 | 0.80 | 0.83 | 0.85 | 0.93 | 0.73 | 0.85 | | | | | | |
| **Black Carbon (BC) mass** | | | | | | | | | | | | | | |
| **5-6km** | 1.09 | 5.01 | 3.30 | 12.59 | 1.21 | 2.15 | 4.16 | 1.84 | 0.59 | 1.78 | 1.75 | | | |
| **4-5km** | 0.84 | 1.21 | 0.47 | 1.20 | 0.50 | 1.28 | 0.50 | 1.40 | 0.39 | 0.78 | 0.48 | | | |
| **3-4km** | 0.82 | 0.84 | 0.83 | 0.79 | 0.85 | 0.93 | 1.11 | 1.25 | 0.55 | 0.87 | 0.77 | | | |
| **2-3km** | 1.04 | 0.90 | 0.97 | 1.28 | 1.79 | 0.90 | 1.11 | 1.81 | 0.86 | 0.70 | 0.72 | | | |
| **1-2km** | 0.89 | 0.79 | 0.95 | 0.90 | 2.60 | 2.57 | 2.22 | 1.87 | 1.05 | 0.91 | 1.23 | | | |
| **0-1km** | 1.68 | 0.88 | 1.17 | 1.03 | 3.48 | 3.09 | 1.06 | 4.82 | 1.77 | 1.08 | 0.68 | | | |
| **Organic Aerosol (OA) mass** | | | | | | | | | | | | | | |
| **5-6km** | 0.71 | 2.50 | 10.08 | 14.56 | 0.81 | 2.37 | 7.02 | 2.80 | 0.37 | 0.73 | 2.27 | | | |
| **4-5km** | 0.71 | 1.25 | 2.28 | 1.58 | 0.60 | 1.82 | 3.59 | 2.86 | 0.51 | 0.99 | 1.32 | | | |
| **3-4km** | 0.84 | 0.77 | 1.24 | 1.36 | 1.37 | 1.33 | 2.69 | 3.39 | 0.68 | 0.82 | 1.17 | | | |
| **2-3km** | 1.09 | 0.89 | 1.23 | 2.61 | 3.31 | 1.58 | 2.28 | 6.49 | 0.98 | 0.87 | 0.89 | | | |
| **1-2km** | 0.96 | 0.89 | 0.92 | 4.46 | 4.48 | 4.38 | 3.05 | 14.63 | 1.46 | 1.47 | 1.20 | | | |
| **0-1km** | 1.35 | 0.79 | 2.02 | 6.40 | 6.71 | 7.29 | 1.92 | 45.33 | 2.00 | 1.94 | 0.81 | | | |





| | Aerosol light extinction ($\sigma_{ep}$) | | | | | | | | *dry* | amb | *dry* | amb | *dry* | amb | | | |
|---|---|---|---|---|---|---|---|---|---|---|---|---|---|---|---|---|---|
| **5-6km** | 0.88 | 0.89 | 3.30 | 7.12 | 0.29 | 0.68 | 1.71 | 0.80 | *0.09* | 0.12 | *0.18* | 0.24 | *0.57* | 0.73 | 0.34 | 0.35 | 0.76 |
| **4-5km** | 0.64 | 0.70 | 0.49 | 0.83 | 0.32 | 0.69 | 0.28 | 0.87 | *0.18* | 0.27 | *0.37* | 0.57 | *0.24* | 0.41 | 0.52 | 0.65 | 0.90 |
| **3-4km** | 0.70 | 0.66 | 0.63 | 0.57 | 0.57 | 0.57 | 0.72 | 0.89 | *0.31* | 0.44 | *0.40* | 0.54 | *0.40* | 0.69 | 0.70 | 0.74 | 1.24 |
| **2-3km** | 1.08 | 0.77 | 0.90 | 1.16 | 1.11 | 0.59 | 0.92 | 1.25 | *0.52* | 0.69 | *0.43* | 0.56 | *0.54* | 0.73 | 1.09 | 0.80 | 2.00 |
| **1-2km** | 1.13 | 1.01 | 0.92 | 1.34 | 2.03 | 2.24 | 1.64 | 1.89 | *0.72* | 1.14 | *0.69* | 0.85 | *0.74* | 1.23 | 1.41 | 1.17 | 2.80 |
| **0-1km** | 1.10 | 1.18 | 0.59 | 1.63 | 3.46 | 4.16 | 1.62 | 5.34 | *0.97* | 4.22 | *0.89* | 3.34 | *0.51* | 1.81 | 6.81 | 4.66 | 2.32 |


meters per week; given the biomass burning smoke is largely advected over the ocean at altitudes >2 km, dry scavenging rates also will be negligible. The vertical position of the plume and how it changes with transport is therefore dominated by overall atmospheric convection and subsidence.

BC and OA are the primary constituents of biomass burning aerosol, so their distribution is a direct measure of the smoke plume intensity and location. On average, the core of the plume is centered at higher altitudes moving towards the edges of the plume (i.e. the south end of the Meridional1 transect and southeast end of the Diagonal transect in 2016, and the north end of the Meridional2 transect in 2017 and 2018) and it covers a broader vertical extent towards the core of the plume (**Figures 4, 5, S.6** and **S.7**). In 2016 in particular this tendency is not captured by any of the models, which have a vertically 960 broader plume across all gridboxes and, in the case of GEOS and UM-UKCA especially, place the plume core at too low an altitude, though the plume top height is nonetheless captured properly in some cases (Shinozuka et al., 2020).

This overly vertically diffuse plume in WRF-CAM5 is consistent with underestimates in BC concentrations in the core of the plume (3-4 km altitude in 2016 and 2018; 3.5-5km in 2017) (**Figure 7**), and over-estimates above and below this. The 965 pattern of bias is similar for GEOS (**Figure 8**), except that it does a better job of reproducing BC concentrations aloft at the southern end of the 2016 Meridional1 transect. UM-UKCA has a significant low bias in BC at plume altitudes in 2016 and a smaller low bias in 2017 (**Figure 9**). In 2016, as with GEOS, the fact that the plume is too low in altitude results in greater (more negative) biases at the highest altitude, but a low bias below this, decreasing from 5 km to 2 km altitude. UM-UKCA also over-estimates the amount of BC (i.e. biomass smoke) that mixes into the marine boundary layer, with consequences for 970 derived forcing through aerosol-cloud interactions.

Above 5 km the observed BC and OA concentrations effectively go to zero (**Figures S.6** and **S.7**), whereas for all models aerosol concentrations taper off more slowly, possibly due to coarse vertical resolution at these heights (e.g., WRF-CAM5 resolution is ~500 m at 6 km). This produces very high modeled-to-observed ratios for the 5-6 km altitude bin. However, this 975 bias will have little effect on column aerosol mass, and therefore aerosol forcing, because concentrations are so low. More









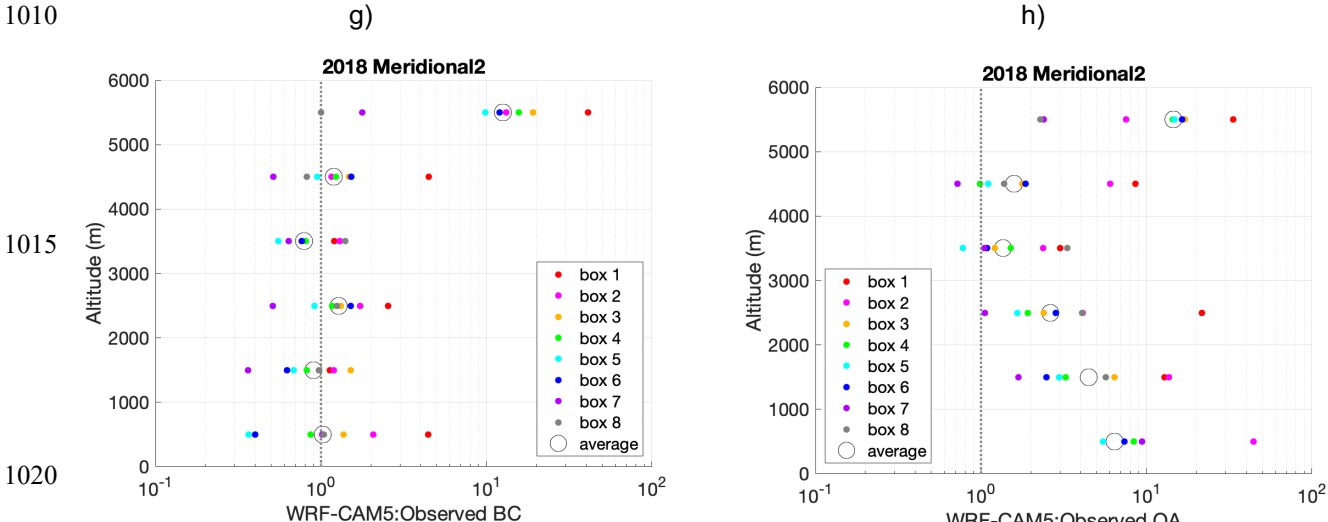

Figure 7. Ratio of modeled to observed BC mass (panels a, c, e and g) and OA mass (panels b, d, f and h) from the WRF-CAM5 model for the Diagonal transect in 2016 (a and b), the Meridional1 transect in 2016 (c and d), the Meridional2 transect in 2017 (e and f) and 2018 (g and h).

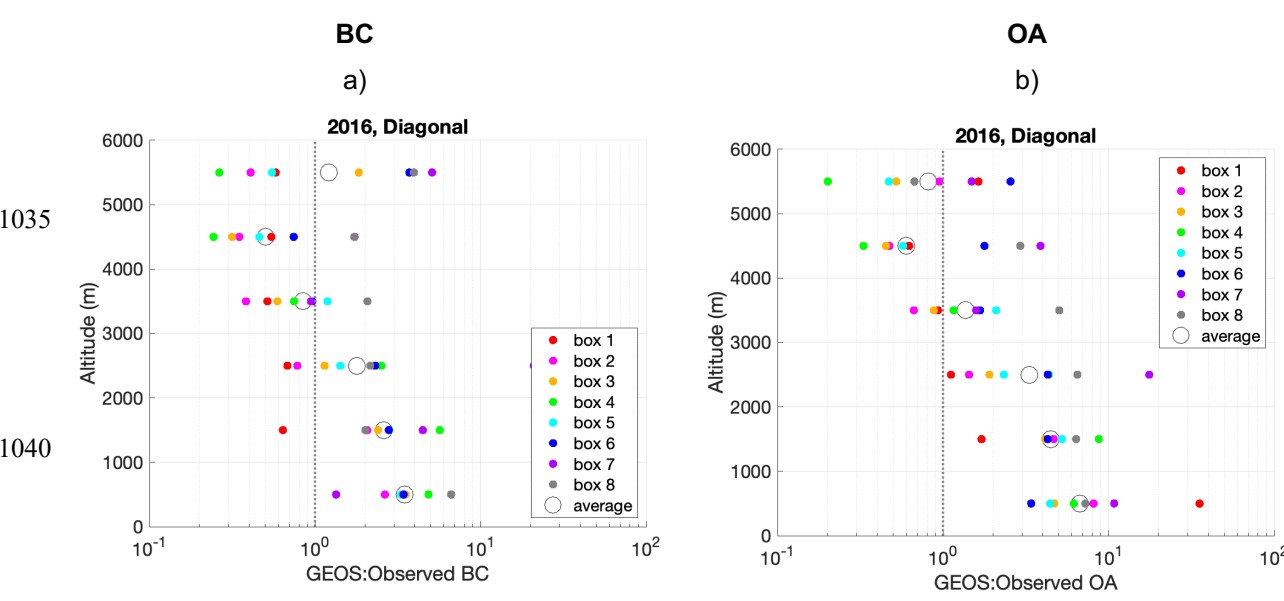





**Figure 8.** As in Figure 7, but for the GEOS model.





**BC**

**OA**

a)

b)

c)

d)

e)

f)










**Figure 9.** As in Figure 7, but for the UM-UKCA model, for 2016 and 2017 only.

consequential is the high bias in modeled BC and OA in the 2-3 km altitude bin. This high bias is particular pronounced for
GEOS for the 2016 comparison transects. The tendency of GEOS to place aerosol too low in altitude can also be seen in the
large high bias in boundary layer (~0-2 km) BC and OA concentrations (**Figure 8**), as reported for the 2016 campaign by
Shinozuka et al. (2016); as in UM-UKCA, this would lead to an overestimate in modeled forcing through aerosol-cloud
interactions.


The marine environment can be a source of OA, but is only a small component of accumulation mode aerosol in the sub-
tropics (Shank et al., 2012; Twohey et al., 2013), and in the models included here the ocean is not a source of OA.
Additionally, there is no marine source for BC. Thus, the high bias in OA and BC concentrations below 2km in the UM-
UKCA and GOES simulations is a clear indication the model is mixing too much biomass burning smoke into the boundary
layer, and therefore into low marine clouds.

### 4.2.3 Light extinction

The comparison between modeled and observed $\sigma_{ep}$ is complicated by the fact that $\sigma_{ep}$ from the in-situ measurements is at
low RH (typically <40%), whereas the models report $\sigma_{ep}$ at ambient RH. An exception is the UM-UKCA model, which
provides both dry and ambient-RH $\sigma_{ep}$. The disparity between low-RH and ambient-RH $\sigma_{ep}$ is expected to be large in the
boundary layer, where RH is generally above 75-80%. Sea salt can be a significant component of boundary-layer aerosol
and, in addition to being very hygroscopic (Tang et al., 1997; Niedermeier et al., 2008), much of it is in the aerosol coarse
mode, which would have been under-sampled by the P-3 aircraft aerosol inlet. Given these issues, the fact that the smoke
resides largely above the boundary layer (e.g., Das et al., 2017), and that the focus of this analysis is on comparisons relevant
to the direct aerosol radiative effect by biomass burning aerosol, our discussion will focus on the comparison of $\sigma_{ep}$ at
altitudes above 2km.

The effect of humification on light scattering is estimated using in-situ measurements of low (<40%) and high (~85%) RH
530 nm light scattering made on the P-3 aircraft in 2016 and 2018. Instrumental problems in 2017 precluded estimates for
that year. The growth of light scattering is parameterized by fitting an exponential function to the measured low- and high-
1140 RH values of $\sigma_{sp}$ versus the RH of the measurements, using the exponent gamma ($\gamma$) as the metric for hygroscopicity
(e.g.,Kasten, 1969; Burgos et al., 2019). Using all data within a given campaign year, $\gamma$ in the plume averages 0.62±0.05 in
2016 and 0.68±0.05 in 2018. This is quite a bit higher than $\gamma$ for biomass burning smoke from previous measurements (e.g.,
Kotchenruther and Hobbs, 1998; Titos et al., 2016). Evaluating the hygroscopicity measurements is beyond the scope of this





paper, but the estimates presented here should be viewed with this in mind. The derived values of $\gamma$ are used directly to calculate the approximate scale factor, f(RH), to convert the low-RH measured values of $\sigma_{sp}$ to ambient RH $\sigma_{sp}$. Values of f(RH) are calculated for the mean±1-sigma observed ambient RH in each comparison transect gridbox within 250m resolution altitude bins from 2km to 6km (**Figure S.8**). For 2017, $\gamma$ from 2018 (0.68) is used in this calculation since the comparison in these two years both cover the same Meridional2 transect.

In 2016, Shinozuka et al., (2020) estimated that f(RH) was less than a factor of 1.2 for 90% of the free troposphere aerosol measurements across the campaign. Estimates for the comparison gridboxes included here are consistent with this (**Figure S.8**), but also show that f(RH) for the 2016 Diagonal gridboxes 3-6 f(RH) was often 1.5 or greater in the 4-5.5 km altitude range. Gridboxes farther north in the 2016 Meridional1 transect also were more humid, with f(RH) for gridboxes 1 and 2 averaging approximately 1.3-1.5, and f(RH) in gridbox 3 exceeding a factor of two in the 4-5km altitude range.

In both 2016 and 2017 RH at plume altitudes was generally <60% (**Figure 6**). In 2017, RH in the plume tended to decrease from north (gridbox 1) to south (gridbox 8) across the Meridional2 transect. The humidification factor f(RH) is accordingly estimated to decrease from typical values of 1.3-1.5 towards the north end of the transect to 1.0-1.2 at the south end (**Figure S.8**). In 2017, f(RH) is again slightly higher at 4-5 km altitude, and the humidity and f(RH) more variable than in the lower part of the plume. This is consistent with the fact that mid-level clouds were intermittently observed within and upwind of this transect.

In 2018, the RH in the plume was greater than in 2016 or 2017 (**Figure 6**). It was still generally 60% or lower in Meridional2 gridboxes 6-8, with f(RH) usually <1.3. North of this, in gridboxes 3-5, RH at 2-4km was closer to 60% so f(RH) is more typically 1.3-1.5. In gridboxes 1 and 2 RH was closer to 80% in the free troposphere. For these gridboxes, f(RH) was almost always greater than 1.5 and could exceed a factor of 2 (**Figure S.8**).

This analysis estimates the effect of humidification on light scattering only, not on light absorption. Since scattering dominates extinction, f(RH) for $\sigma_{sp}$ nonetheless provides a good estimate of the impact of humidification on $\sigma_{ep}$. Based on this analysis, the in-situ, low-RH values of $\sigma_{ep}$ are expected to typically be 20-50% lower than the modeled and HSRL-2 measured ambient-RH values of $\sigma_{ep}$, all else being equal. Instead, the modeled values of ambient-RH $\sigma_{ep}$ in the plume are generally lower than both the dry in-situ values and the ambient-RH HSRL-2 values.

As for OA and BC, the observed $\sigma_{ep}$ profiles are at higher altitude and less vertically diffuse than in the models (**Figures 4 and 5**). In the in-situ measurements in particular $\sigma_{ep}$ increases more rapidly with altitude at the bottom of the plume and decreases more rapidly with altitude at the top of the plume than does the WRF-CAM5 modeled extinction. This is most

**In-situ**                                                                 **HSRL-2**












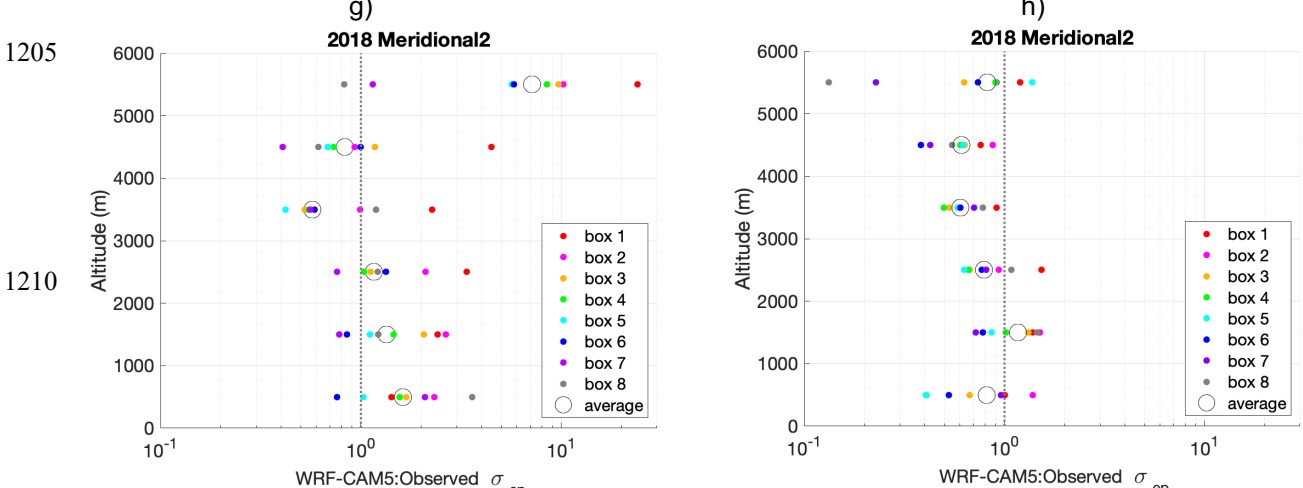

**Figure 10.** Ratio of modeled to in-situ dry (left column) or HSRL-2 ambient-RH (right column) mid-visible $\sigma_{ep}$ to modeled ambient-RH $\sigma_{ep}$ for the WRF-CAM5 model, for a,b) the Diagonal transect in 2016, c,d) the Meridional1 transect in 2016, e,f) the Meridional2 transect in 2017 and g,h) the Meridional2 transect in 2018. Note the variable scale of the abscissa.

pronounced in the gridboxes with higher-concentration and/or, more well-defined plumes (e.g., **Figure 4a**). In addition, the observed plume is centered at a higher altitude than in the models. This leads to under-estimates in modeled $\sigma_{ep}$ in WRF-CAM5, relative to in-situ values, in the core of the plume (3-4 km altitude) of about 30-35% in 2016, 10% in 2017 and 15% in 2018 (**Table 1**; **Figure 10**). Below and above the core of the plume (the 2-3 km and 4-5 km bins), the modeled-to-in-situ-observed ratio in $\sigma_{ep}$ is closer to or greater than 1.0. In 2017 and 2018, WRF-CAM5 modeled extinction at 5-6km is more than 4 times (2017) and 9 times (2018) greater than observed in-situ, but this is because $\sigma_{ep}$ is measured to be near zero above 5 km in most gridboxes.

Biases in the model when compared to $\sigma_{ep}$ from HSRL-2 follow a similar but less consistent pattern (**Figure 5** and **Table 2**). Again, the measurements show a plume core that is centered at a higher altitude and is less vertically diffuse than in the models, especially in 2016 (**Figure 5**). The model mean low bias referenced to the HSRL-2 measurements is generally greater than the mean low bias compared to in-situ observed $\sigma_{ep}$ (compare **Tables 1** and **2**), consistent with the former being at ambient RH and the latter dry $\sigma_{ep}$. Except in the 2018 Meridional2 transect, WRF-CAM5 $\sigma_{ep}$ is generally 30-40% lower than measured by HSRL-2.

Notable in comparing **Figures 4** and **5** is that in 2016, when the HSRL-2 was on board the ER2 and so had retrievals to >6km altitude, the top of the plume extends to higher altitudes than covered by the in-situ measurements. In the latter, $\sigma_{ep}$ drops to near zero above 6 km in most comparison gridboxes; in the HSRL-2 retrievals, $\sigma_{ep}$ above 5500m is usually still





>50Mm$^{-1}$. The in-situ and HSRL-2 measurements were not coincident, so this difference could simply reflect different sampling, but the consistency of this feature across multiple comparison transects makes this seem unlikely. Relative

humidity often increased above about 4 km (**Figure 6**), with humidification often amplifying $\sigma_{ep}$ by a factor of 1.5 or more (**Figure S.8**). While this cannot fully account for the very large difference in $\sigma_{ep}$ in the models versus that observed in-situ, or all of the difference between the plume top behavior between the in-situ and HSRL-2 measurements, humidification differences could be contributing.

The net effect of these altitude-dependent biases in $\sigma_{ep}$ is that WRF-CAM5 under-estimates plume AOD, with Shinozuka et al. (2020) calculating a low bias of 10-30% in AOD for the 2016 campaign. Here, the modeled ambient-RH $\sigma_{ep}$ is typically 70-80% of the in-situ measured dry $\sigma_{ep}$ (with considerable variability). Accounting for the difference in humidity of the in-situ measurements (i.e. scaling in-situ $\sigma_{ep}$ by 1.2-1.5) would make the WRF-CAM5 $\sigma_{ep}$ in the plume to only about 50-70% of the observed average. This is not far from the observed ratios of WRF-CAM5 to HSRL-2 observed ambient-RH $\sigma_{ep}$ (**Table**

**2**).

**Table 2.** As in Table 1, but for retrieved values of $\sigma_{ep}$ only from the HSRL-2 lidar on board the ER-2 aircraft in 2016 and the P-3 aircraft in 2017 and 2018. Both modeled and observed values are at ambient RH, but the modeled values are 550nm and the retrieved values are at 532nm.

| | WRF-CAM5 | | | | | GEOS | | | | | UM-UKCA | | | |
|---|---|---|---|---|---|---|---|---|---|---|---|---|---|---|
| | 2016 Diag | 2016 Mer1 | 2016 Zon | 2017 Mer2 | 2018 Mer2 | 2016 Diag | 2016 Mer1 | 2016 Zon | 2017 Mer2 | 2018 Mer2 | 2016 Diag | 2016 Mer1 | 2016 Zon | 2017 Mer2 |
| **Aerosol light extinction ($\sigma_{ep}$)** | | | | | | | | | | | | | | |
| **5-6km** | 0.18 | 0.34 | 0.41 | -- | 0.82 | 0.08 | 0.17 | 0.17 | -- | 0.24 | 0.09 | 0.12 | 0.21 | -- |
| **4-5km** | 0.40 | 0.52 | 0.34 | -- | 0.61 | 0.22 | 0.41 | 0.25 | 0.57 | 0.50 | 0.32 | 0.65 | 0.45 | 0.39 |
| **3-4km** | 0.63 | 0.66 | 0.52 | 1.06 | 0.60 | 0.55 | 0.49 | 0.48 | 0.78 | 0.79 | 0.54 | 0.70 | 0.54 | 0.77 |
| **2-3km** | 0.64 | 0.66 | 0.71 | 0.86 | 0.79 | 0.65 | 0.67 | 0.79 | 0.93 | 1.00 | 0.74 | 0.72 | 0.76 | 0.60 |
| **1-2km** | 0.26 | 0.33 | 0.64 | 0.57 | 1.17 | 0.53 | 0.67 | 1.12 | 0.94 | 1.57 | 0.48 | 0.41 | 0.83 | 0.62 |
| **0-1km** | 0.35 | 0.50 | 0.19 | 0.31 | 0.82 | 1.03 | 1.27 | 1.33 | 0.87 | 2.07 | 1.02 | 0.92 | 0.85 | 1.10 |


Biases in GEOS modeled $\sigma_{ep}$ profiles have a strong vertical gradient in most comparison transects, with generally positive biases below about 2 km; above this the model has a low bias that increases with altitude (**Figure 11** and **Table 1**). The low bias in GEOS $\sigma_{ep}$ in the plume is stronger to the west (northwest end of 2016 Diagonal transect, **Figure 4a**, and west end of the Zonal transect, **Figure 5c&d**) than to the east, indicating an issue with insufficient transport of aerosol in the Southern

African Easterly jet and/or too-short aerosol lifetimes. The low bias in GEOS $\sigma_{ep}$ relative to both the in-situ and HSRL-2 measurements is smaller for August 2017 than for September 2016, and smaller again for October 2018. (**Tables 1** and **2**).




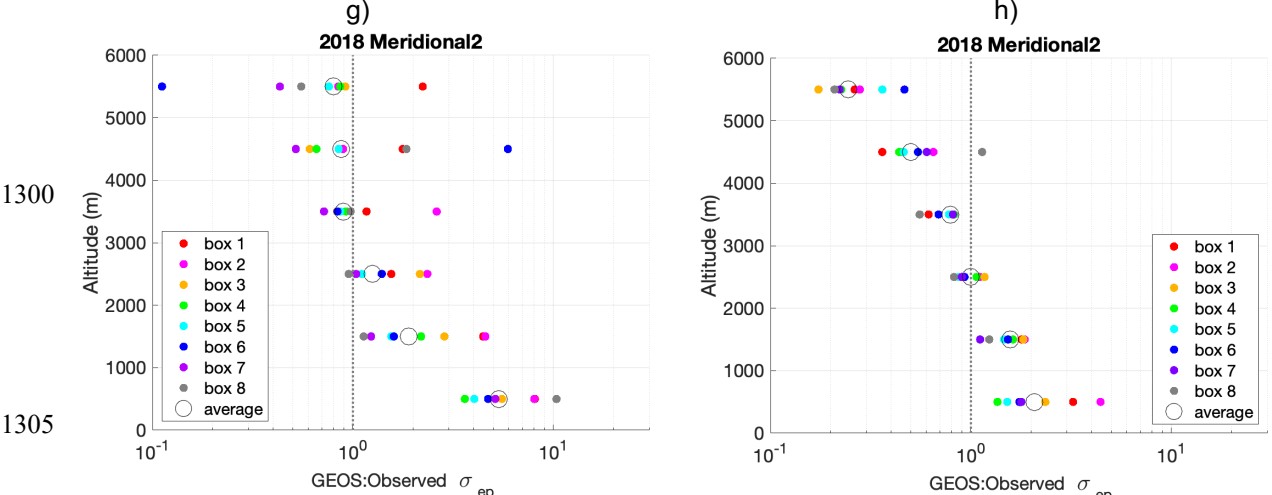

**Figure 11.** As in Figure 8, but for the GEOS model.

The bias also has a less consistent dependence on altitude in 2017 (**Figure 11**). In 2018, the higher ambient RH could be compensating for some of the low bias in dry aerosol $\sigma_{ep}$. Overall, it is clear that GEOS underestimates $\sigma_{ep}$ in the plume and centers the plume at too low an altitude (**Figures 4** and **5**), with a net impact of under-estimating the above-cloud AOD. Shinozuka et al. estimated a greater low bias in GEOS AOD (30-50%) than in WRF-CAM5 (10-30%). As for WRF-CAM5, accounting for humidification in the in-situ observations would increase the estimated bias in GEOS, to greater than a factor of two. This is somewhat surprising, given that GOES assimilates satellite-retrieved AOD every three hours (Albayrak et al., 2013).

For both transects in 2016, ALADIN-modeled $\sigma_{ep}$ is underestimated at the core of the plume and the modeled plume is too vertically diffuse (**Figure 4**). Also apparent is that the model increasingly places the plume at too low an altitude moving from southeast to northwest in the Diagonal1 transect, consistent with too much subsidence in the model with aerosol transport, as in GEOS (Das et al., 2017). In the 2016 Diagonal transect this produces significant low model biases for 3-5 km altitude and high biases for 2-3 km and 5-6 km (**Figure 12**; **Table 1**), much as for the WRF-CAM5 model. In 2017, ALADIN biases in $\sigma_{ep}$ along the Meridional2 transect don't have a significant altitude dependence. There is, however, a tendency for the model to over-estimate $\sigma_{ep}$ at the north end of this transect and under-estimate it at the south end, very possibly due to humidification amplifying $\sigma_{ep}$ by about a factor of two for the northern gridboxes, but only by a factor of ~1.1-1.4 at the southern end (**Figure S.8**). Statistics from the ALADIN model are not available for comparison to the HSRL-2 retrieved $\sigma_{ep}$. Notably, the ALADIN simulations were run using fixed 2014 GFED emissions. Central and southern African biomass burning emissions in 2014 were not particularly different from the 2001-2013 climatological average (Kaiser and Van der Werf, 2015). While not a direct measure of emissions, AOD over the SE Atlantic was in both September 2016 and August 2017 than the 2003-2018 average (Redemann et al., 2021), consistent with lower emissions in these months and



**In-situ**

a)

**2016 Diagonal**

**HSRL-2**

b)

**2016 Meridional1**

c)

**2017 Meridional2**

**Figure 12.** As in Figure 8, but for the ALADIN model, for observed in-situ values from the P-3 in 2016 and 2017 only, for the a) 2016 Diagonal transect, b) 2016 Meridional1 transect and c) 2017 Meridional1 transect.

years than on average. If the ALADIN simulations used emissions for these months, modeled $\sigma_{ep}$ may have been smaller, with greater low biases in comparison to the observations.

For the UM-UKCA model dry as well as ambient-RH $\sigma_{ep}$ values were reported, allowing for a more robust comparison to the measured low-RH $\sigma_{ep}$ and a rough comparison of observed versus modeled humidification factors. At all altitudes above 2 km, the model under-predicts dry aerosol $\sigma_{ep}$ significantly across all comparison transects in 2016 and 2017 (**Figure 13** and **Table 1**). This low bias increases systematically with altitude from 2 km to 5 km. The gridbox-mean dry $\sigma_{ep}$ is typically a





**In-situ**

**HSRL-2**












**Figure 13.** As in Figure 8, but for the UM-UKCA model, for 2016 and 2017 only. For the UM-UKCA
model statistics for both dry and ambient-RH $\sigma_{ep}$ were provided. The ratio of both to the observed (dry) $\sigma_{ep}$
are shown (dots, dry; squares, ambient-RH) for the comparisons to the P-3 in-situ data (panels a, c, e).

factor of two to three lower in the 3-4km and 4-5km altitude bins in the model than observed in-situ (**Figure 13** and **Table
1**). The altitude dependence in the model biases again results from the modeled plume being too vertically diffuse, and the
plume core too low in altitude.

Even with humidification added, the modeled extinction is lower than the observed dry $\sigma_{ep}$, despite the fact that the model
aerosol appears to be too hygroscopic. In the modeled comparison transect averages, $\sigma_{ep}$ in the 2-5km altitude range is a
factor of 1.4 higher at ambient RH than for the dry aerosol; for 2017 it's a factor of 1.5 higher. These are somewhat higher
than expected from our analysis from the in-situ observations, where f(RH) at these altitudes averaged 1.2 in both 2016 and
2017 (**Figure S.8**). The significant contribution of humidification to $\sigma_{ep}$ also manifests in the fact that the modeled ambient-
RH $\sigma_{ep}$ is typically 55-75% of that observed from the HSRL-2 (**Figure 13** and **Table 2**) – a much smaller low bias than the
factor of 2-3 model underestimate in dry $\sigma_{ep}$.

The UM-UKCA low biases in dry $\sigma_{ep}$ are much greater than the model low biases in OA and BC, indicating that the model
has a low bias in biomass burning aerosol mass extinction efficiency as well as mass.  There could also be simply less total
aerosol mass (e.g. of components other than OA and BC, such as sulfate) in the UM-UKCA model than in reality.

### 4.3 Biases in aerosol intensive optical properties

Model biases in aerosol constituent component masses (BC, OA) and $\sigma_{ep}$ can arise from a combination of biases in the
emissions, transport, deposition and (for OA) in-atmosphere production and loss from the aerosol phase of the biomass
plume aerosol, which will clearly affect the magnitude of the calculated direct aerosol radiative effect. As noted earlier, the
aerosol intensive optical properties, in particular the SSA, will affect the sign of the aerosol DARE. The SAE connects the
aerosol mass and extinction through the aerosol size, which is directly related to its aerosol mass scattering efficiency. The
SAE and AAE, combined with mid-visible $\sigma_{ep}$, give the wavelength-dependence of SSA (Russell et al., 2010).

### 4.3.1 Single scatter albedo (SSA)

Observed and modeled SSAs differ in two respects: in their absolute value and in their variation with altitude. In Sept. 2016
the observed SSA increases with altitude within the biomass burning plume along both the Diagonal and Meridonal1
transects (**Figure 14**), generally increasing from 0.82-0.84 at the bottom of the plume to 0.86-0.88 at the plume top. In















**Figure 14.** Profiles of single scatter albedo (SSA) as observed (at low RH and 530nm) and modeled (at ambient RH and, for UM-UKCA, low RH, at 550nm) for the a) Diagonal transect in Sept. 2016, b) Meridional1 transect in Sept. 2016, c) Meridional2 transect in Aug. 2017 and d) Meridional2 transect in Oct. 2018. Shown are means and standard deviations across all times and locations within a comparison gridbox when in-situ measurements were made.



August 2017 SSA spanned a similar range as in 2016 (**Figure 15**), but there is no significant gradient in SSA with altitude in the northern four Meridional2 gridboxes, and only a slight indication of an increase in SSA with altitude towards the southern half of the plume. There is also no vertical gradient in SSA in the northern-most three Meridional2 gridboxes in Oct. 2018, and SSA is overall higher than in Sept., 2016 and Aug., 2017. A vertical gradient is apparent in the southern five

Meridional2 gridboxes in 2018, where SSA increases from 0.86-0.89 at plume bottom to 0.90-0.92 at plume top. Notably, gridboxes 4-7 are located well within the biomass plume, whereas the north end of this transect was often outside or on the edges of the plume. The vertical gradient in SSA has been associated with a gradient in aerosol composition (Redemann et al., 2021). Accounting for this gradient is important in determining the direct aerosol radiative effect because it is the extinction-weighted column SSA, combined with below-plume albedo, that dictates the sign of the direct forcing.


WRF-CAM5 produces little to no associated gradient in SSA with altitude (**Figure 14**), showing increases in SSA only at the very top of the plume in some comparison gridboxes. Within the plume, SSA encompasses quite a small range in the model, almost always being 0.82-0.84 in the Sept. 2016 transects and the Aug. 2017 transect (**Figure 15**). This difference in vertical gradient explains why Shinozuka et al. (2020) find greater low biases in the 3-6km altitude column SSA than in the lower

free troposphere column SSA. As in the observations, in October 2018 the WRF-CAM5 SSA is slightly higher than in the other years, generally 0.84-0.86. WRF-CAM5 does not consider OA and BC aging and primary OA hygroscopicity is low (0.1), which is consistent with the small ranges shown for ambient SSA.

GEOS similarly has little gradient in SSA with altitude within the plume, and where it does the tendency is for SSA to

decrease with altitude, particularly towards the plume top, where it diverges from the observed, increasing dry aerosol SSA. Mean plume SSA values on average are similar in GEOS and WRF-CAM5, but SSA is about twice as variable in GEOS. This larger range in SSA doesn't show any apparent spatial pattern, other than somewhat higher SSA in the northern-most Meridional2 gridbox in both 2017 and 2018. This gridbox tended to be on the northern edge, or out of, the main biomass burning plume. Aging of OC GEOS could be creating more hygroscopic aerosol with time, which in turn would increase the

variability in ambient SSA through differences in water take-up.

In the UM-UKCA simulations SSA also decreases significantly towards the top of the plume in some comparison gridboxes. In most cases this is true for both the dry and ambient-RH aerosol SSA, so this appears to be driven by a change in aerosol composition or size not by, e.g., a decrease in scattering due to a decrease in RH. ALADIN is the only model of the four

where the SSA (which is at ambient RH) has a similar gradient with altitude in the plume to that observed, though the modeled SSA is consistently higher than that observed. SSA in ALADIN includes a dependence on aerosol aging and RH (Mallet et al., 2019), and thus it is not clear if the trends shown are a response to the larger RH on the top the plume or would still be present under dry conditions.







**Figure 15.** Histograms of SSA from the 2-5km altitude range only, as measured in-situ from the P-3 and as modeled using WRF-CAM5, GEOS-5, UM-UKCA and ALADIN for all samples along a) the Diagonal transect in 2016, b) the Meridional1 transect in 2016, and along the Meridional2 transect in c) 2017 and d) 2018. For the UM-UKCA model, values are shown for both ambient-RH aerosol (filled bars) and dry aerosol (unfilled bars).





The UM-UKCA ambient-RH SSA values are within 0.02 of the values measured in-situ at most altitudes in the plume

(**Figure 14**), but the dry-aerosol SSA from UM-UKCA is significantly lower (typically 0.77-0.83) than both the modeled ambient-RH values (typically 0.84-0.88) and the dry in-situ values (**Figures 14** and **15**). This difference between the SSA of the dry and ambient-RH aerosol results from the significant increase in extinction with RH (Section 4.3.2).

In reality, it's likely that humidification affects both scattering and absorption. However, the former has been well-quantified

observationally, whereas the latter has not and is therefore highly uncertain (e.g., Bond et al., 2013; Zhou et al., 2020). The models include the effect of humidification on SSA by accounting for the impact of water on the aerosol indices of refraction (WRF-CAM5, GEOS, UM-UKCA) or by parameterizing the effect of RH on SSA (ALADIN; Mallet et al., 2017; 2019). In all four models, the result is an increase in SSA with humidification.

Despite this, the modeled ambient-RH SSA at plume altitudes is generally lower than the observed dry-aerosol SSA in both the WRF-CAM5 and GEOS models. Thus, the dry aerosol SSA in these models has even a greater low bias than indicated by **Figures 14** and **15**. Humidity in the plume was somewhat higher in 2017 and, in particular, 2018 than in 2016 (**Figure 6**), increasing modeled SSA and moving the modeled (ambient-RH) and observed (dry) SSA values in closer alignment on average. In contrast, for the two 2016 transects the ALADIN ambient-RH SSA is almost always higher than the observed dry

SSA, with typical differences of 0.02-0.04. Further, the differences are smaller in the 2017 Meridional2 transect, particularly towards the south, despite the higher ambient RH.

As discussed below (Section 5.0), the relative biases in OA and BC indicate that the models have a higher OA:BC ratio than is observed. For given indices of refraction for these components, a higher OA:BC ratio would increase SSA in the models

(since they don't include organic aerosol "brown" carbon absorption). Thus, the model OA:BC ratio does not explain the low bias in SSA in the WRF-CAM5 and GEOS models. Biases in model aerosol component indices of refraction, incorrect representation of the impacts of internal mixing on indices of refraction, and the influence of aerosol components other than BC and OA could all be contributing to the observed modeled biases in SSA.

An earlier study using data from the ORACLES 2016 field season compared SSA derived from the in-situ measurements used here and from three remote sensing instruments (Pistone et al., 2019): a spectral radiometer (SSFR) in combination with sun photometer derived AOD, a hyperspectral Sun photometer and sky radiometer (4STAR) in combination with SSFR derived scene albedo, and an imaging polarimeter (AirMSPI). The SSFR and 4STAR instruments were deployed on the NASA P-3 aircraft along with the in-situ instruments; the AirMSPI instrument was mounted on the NASA ER-2 high-

altitude aircraft, which overflew the P-3 at least once on coincident flight days (See Pistone et al, 2019 for more detail). The remote sensing instruments, like the models, all derive SSA at ambient RH. At 530nm, the average distribution of SSA from the in-situ instruments was higher than the 4STAR SSA by (on average) 0.01-0.02 (with 10-90 percentile ranges of 0.07 in





each), but was generally lower than the AirMSPI SSA, with differences of 0.03, with less variability overall (0.03 in 10-90 percentiles) compared with the P-3 instruments. The spread in differences was likely in part due to the full-campaign measurements not being coincident in either time or space, due to the varying measurement techniques. Direct comparison to the SSFR was made in one case study only, and for that case the SSFR 530nm SSA was <0.01 lower than the in-situ SSA (within the instrument uncertainty ranges). These results indicate it is unlikely that SSA from the in-situ instruments is significantly biased high. It also supports the idea that humidification is not significantly influencing SSA over the SE Atlantic, since the in-situ values are at low RH and the remote sensing values are at ambient RH.

### 4.3.2 Scattering Ångström Exponent (SAE)

Whereas SSA varies primarily with aerosol composition, SAE varies primarily with aerosol size: lower values of SAE correspond to large aerosol size for aerosol smaller than approximately 1000 nm dry diameter (Schuster et al., 2006), which is the case for aerosol in the plume for both the observations and all models (Shinozuka et al., 2020). The observed SAE is quite consistently 1.7-1.9 within the plume, with very little vertical variation or difference across the two campaign years (2016 and 2017) where observations are available (**Figure 16**). WRF-CAM5 simulated SAE deviates the furthest from the observed values, with values in the plume generally 0.9-1.3, consistent with the larger aerosol size in WRF-CAM5 than in reality (see Figure 4 in Shinozuka et al., 2020). GOES and UM-UKCA both reproduce the observed SAE quite well, with the exception of a few gridboxes. The GEOS SAE values are typically 0.1-0.2 smaller than observed in the 2016 Meridional1 gridboxes 1-3, and above about 4 km at the north end of the 2017 Meridional2 transect.

In the UM-UKCA model, the dry aerosol SAE is not always significantly greater than the ambient-RH aerosol SAE. This is somewhat surprising since the difference in dry and ambient-RH $\sigma_{ep}$ (**Figure 4**) and SSA (**Figures 14** and **15**) are significant in this model within the boundary layer, indicating significant growth with humidification (Section 4.3.1). The dry aerosol is therefore significantly smaller, and so we would expect it to have more consistently higher SAE.

Both the dry and ambient-RH UM-UKCA values of SAE agree well with the observed SAE, except in the southeast-most 3 gridboxes of the 2016 Diagonal transect, gridbox 7 of the 2016 Meridional1 transect, and above 4 km in the 2017

Meridional2 transect gridboxes, where the UM-UKCA values are higher than observed. As noted above, the UM-UKCA aerosol is only slightly larger than observed, so SAE should also be similar.

Both GEOS and UM-UKCA simulate vertical gradients – but of opposite sign – in SAE, and a gradient is not present in the observations. The ALADIN model also has a less-pronounced decrease in SAE towards the top of the plume and it simulates SAE values that are typically 0.02-0.04 higher than are observed in almost all gridboxes and plume altitudes. This is consistent with the aerosol in ALADIN being smaller than the observed aerosol, but the model uses a bulk-bin scheme so a





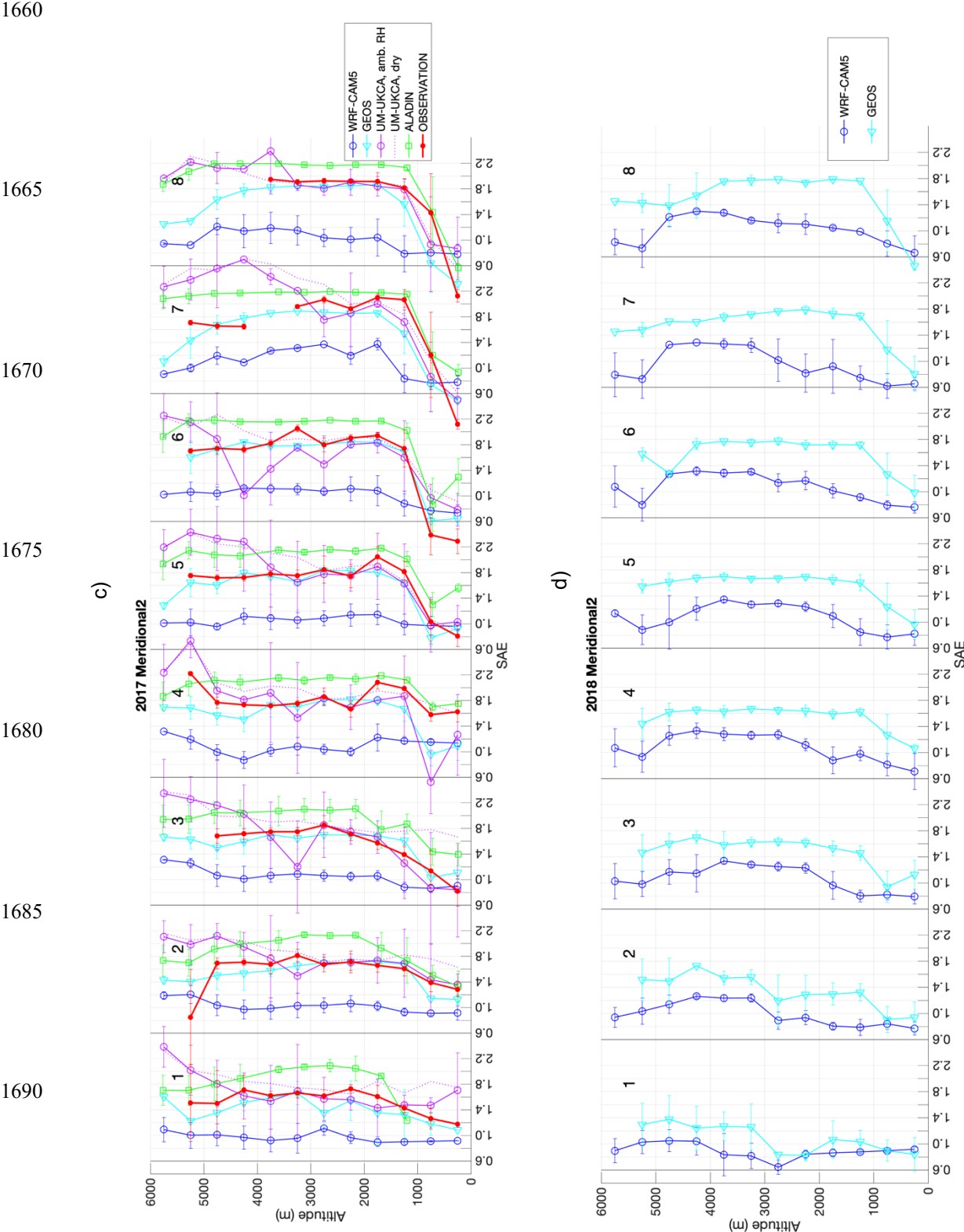

**Figure 16.** As in Figure 12, but for the scattering Ångström exponent (SAE).





mean aerosol size is not available for comparison. The GEOS and ALADIN models do include the effect of aging on aerosol
hygroscopicity, possibly driving the modeled gradients in SAE.

### 4.3.3 Absorption Ångström Exponent (AAE)

As noted above, the in-situ PSAP measurements of $\sigma_{ap}$ at 530, 460 and 660 nm are processed using two sets of correction factors, wavelength-averaged and wavelength-specific (Virkkula, 2010). The two are the nearly identical at 530nm but differ at 460 and 660nm and so produce quite different values of the absorption Ångström exponent, both versions of which are
shown in **Figure 17**. In the 2-5.5 km altitude range the wavelength-independent correction factors yield AAE values of 1.2±0.1 in 2016 and 1.1±0.2 in 2017 and 2018. AAE at these same altitudes calculated with the wavelength-specific correction factors are about 0.3-0.4 higher, on average 1.5±0.1 in 2016 and 1.5±0.2 in 2017 and 2018.

The Pistone et al., (2019) comparison of spectral SSA across different instruments during ORACLES 2016 used SSA
calculated from $\sigma_{ap}$ using the wavelength-averaged correction factor. Using the wavelength-specific correction factors produces higher 470nm absorption, with the result that SSA is lower at both 470nm and 660nm than at 530nm (Pistone et al., 2019). In contrast, SSA derived from absorption using the wavelength-averaged correction factors decreases with wavelength. The latter agrees better with the shape of the spectral SSA from the remote sensing instruments, indicating the lower values of AAE (derived from $\sigma_{ap}$ values using the wavelength-averaged correction) are more likely to be correct.
Values of AAE close to 1 are consistent with the absorption being dominated by black carbon (Bergstrom et al., 2007; Bond et al., 2013), so if these lower values are correct there is likely little brown carbon absorption, as also indicated by other recent studies of biomass smoke in the SE Atlantic (Chylek et al., 2019; Denjean et al., 2020; Taylor et al., 2020). The four models included here do not include absorption by brown carbon, so AAE values are expected to be near 1.

AAE from the WRF-CAM5 and GEOS models are significantly lower than the in-situ wavelength-specific values and slightly but not significantly lower than the wavelength-independent values. In both models AAE in the 2-5.5 km altitude range is 1.1 in 2016 and 2017 and 1.2 in 2018, with standard deviations of <0.05. The UM-UKCA AAE values vary from the observed values (**Figure 17**) but on average are in good agreement with the in-situ AAE derived using the wavelength-specific correction factors.

## 4.4 Biases in cloud fraction and cloud optical thickness

Observed cloud properties are retrieved from satellite observations and so are available from every day of the three field campaign periods and for the Zonal transect as well as for the Diagonal, Meridonal1 and Meridional2 transects. Mean $CF_{warm}$ and geometric mean $COT_{warm}$ from the satellite retrievals are compared to model averages across all daytime hours (**Figure 18**). We treat the SEVIRI-LaRC retrievals as the benchmark for $CF_{warm}$ since these measurements cover the daytime hours,
and the MODIS-ACAERO retrievals as our benchmark for $COT_{warm}$, since it accounts for the effects of absorbing aerosol









**Figure 17.** As in Figure 12, but for the absorption Ångström exponent (AAE).



above the clouds, while acknowledging that any satellite retrievals of clouds may be subject to systematic biases. In
cumulus cloud regions in particular 3-D radiative effects and sub-pixel clear-sky contamination may be biasing the retrieved
values of $COT_{warm}$ low (e.g., Marshak et al., 2006; Kato et al., 2006; Painemal et al., 2013) whereas the coarse pixel
resolution relative to the cloud size could yield an overestimation of $CF_{warm}$ (e.g., Zhao and Di Girolamo, 2006).

The observed $CF_{warm}$ from all three satellite data products is quite high (>60-70%) across almost all comparison transects
gridboxes, except at the southeast end of the Diagonal transect in 2016 (**Figure 18**). Differences in the observation times of
the MODIS and SEVIRI instruments are expected, on average, to lead to $CF_{warm}$ values that are about 1-10% lower for the
two MODIS products (Section 3.2.3 and **Table S.2**), but in our statistics $CF_{warm}$ from MODIS-Standard and MODIS-
ACAERO are not always lower than from SEVIRI-LaRC (**Figure 18**). Differences in the spatial resolution (3km for
SEVIRI; 1km for MODIS), algorithms used for identifying warm clouds, and in the aggregation of statistics (e.g., the L2
datasets for MODIS-ACAERO versus the L3 dataset for MODIS-Standard) could also be producing differences in derived
$CF_{warm}$. Notably, the uncertainty in the "true" $CF_{warm}$ as expressed through the differences between the satellite products is
much lower than the differences between $CF_{warm}$ from the observational datasets and the models.

$CF_{warm}$ in WRF-CAM5 is higher than in all three observational datasets. This is particularly the case in regions of low cloud
fraction, so the gradients in cloud fraction across the transects follow the tendency of the observed gradients but are much
smaller in magnitude. In contrast, the GEOS and ALADIN models both significantly underestimate $CF_{warm}$ in all transects
and almost all gridboxes. $CF_{warm}$ gradients in the ALADIN model also track well the observed gradients, but at much lower
cloud fraction. The GEOS model, in addition to significantly under-estimating $CF_{warm}$, fails to capture the correct gradient in
cloud fraction. In particular, the latitudinal gradient in cloud fraction along the Meridional2 transect in both 2017 and 2018 is
1815 the inverse of that observed. The UM-UKCA model comes closest to the observed cloud fractions, with variable biases
depending on the transect and gridbox. Like WRF-CAM5, UM-UKCA biases in cloud fraction are largest where it fails to
capture spatial gradients in cloud fraction – e.g., along the Meridonal1 transect, at the north end of the Meridional2 transect,
and at the east end of the Zonal transect.

In the SE Atlantic, the large-scale gradients in day-to-day cloud fraction are controlled by a number of different intertwined
factors, including gradients in sea surface temperature (SST) from the Benguela current off the Namibian–Angolan coast and
by lower tropospheric stability (Wood, 2012). Increases in $CF_{warm}$ in the SE Atlantic have been shown to be correlated with
increases in lower tropospheric stability (LTS), surface wind speed and RH at 950 hPa (Fuchs et al., 2018). In the models,
failure to capture the dynamics that drive one or more of these variables could be the cause of incorrect gradients in $CF_{warm}$.
Differences in the representation of small scale turbulent mixing processes and microphysics have been shown to hamper
model skill in representing stratocumulus properties, even when large-scale forcings are fixed (Zhu et al., 2005; Wyant et al.,







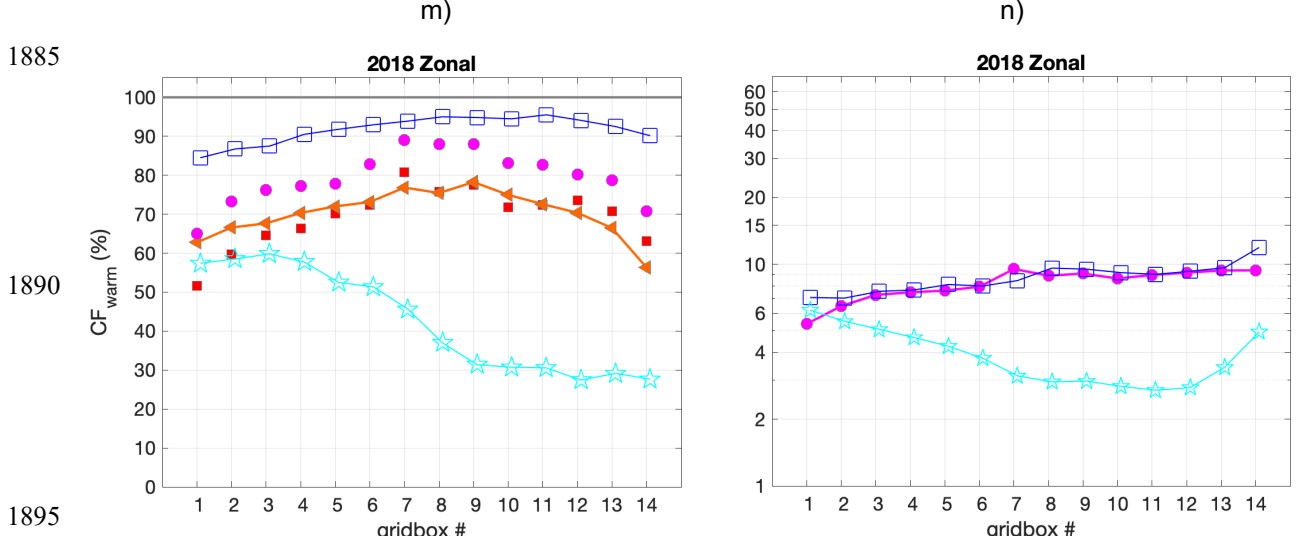

**Figure 18.** Average daytime mean CF$_{warm}$ (left) and COT$_{warm}$ (right) from satellite retrievals and from the models for the comparison gridboxes in the Diagonal transect in 2016 (a & b), Meridional1 transect in 2016 (c & d), Meridonal2 transect in 2017 (e & f), Meridonal2 transect in 2018 (g & h), and Zonal transect in 2016 (i & j), 2017 (k & l) and 2018 (m & n). COT$_{warm}$ is the geometric mean, except for the ALADIN model which shows the median. The SEVIRI-LaRC dataset is used as the reference observed CF$_{warm}$, and the MODIS-ACAERO dataset is used as the reference observed COT$_{warm}$ in the comparison to models.

2007). Accurate prediction of cloud cover is complicated because the relative roles of different large-scale forcings varies across the region (Fuchs et al., 2018).

Disentangling the sources of model biases in COT$_{warm}$ are even more difficult, especially since COT$_{warm}$ may be affected not only by thermodynamic processes but is also very sensitive to cloud microphysical process representation (Wyant et al., 2007) and to aerosol-cloud interactions. As discussed above, the models tend to place the smoke plume at lower altitudes than observed, and therefore to mix more of the plume into the boundary layer and low clouds than is observed. This would lead to higher aerosol loadings in the boundary layer, higher cloud droplet number concentrations, and possibly higher COT$_{warm}$, (assuming cloud liquid water path is not significantly reduced in response; Twomey 1977). An exception is for the ALADIN COT$_{warm}$, as aerosol microphysical effects on clouds were not simulated. Diagnosing the possible magnitude of this effect on modeled COT$_{warm}$ in the other models is beyond the scope of this paper.

The observed (MODIS-ACAERO retrieval) geometric mean COT$_{warm}$ covers a fairly small range, of 8.3±1.8 across all transects and years (**Figure 18**). WRF-CAM5 COT$_{warm}$ values (8.1±1.9) are very similar to the observed values on average





but deviate from the observed values by not capturing spatial gradients. The result is that WRF-CAM5 sometimes over-estimates and sometimes under-estimates COT$_{warm}$. ALADIN generally reproduces the observed COT$_{warm}$ well, except at the western end of the Zonal transect, where COT$_{warm}$ is about twice that observed. GEOS COT$_{warm}$ is both too small and more spatially variable than observed (6.6±3.2); like CF$_{warm}$, the model also doesn't capture the correct spatial gradients in COT$_{warm}$. UM-UKCA, significantly overestimates COT$_{warm}$ across all transects, averaging 32.6±6.4.

It has been noted that global models tend to have a "too few, too bright" bias for low-level marine clouds (Nam et al., 2012). Here, none of the four models fit this paradigm, including the two global models (GEOS and UM-UKCA). The regional WRF-CAM5 model has too many clouds, but the clouds are of about the right brightness; the regional ALADIN model has too few clouds that are generally not bright enough, but sometimes too bright; clouds in the global GEOS model are both too few and not bright enough; and the global UM-UKCA model has too many clouds that are much too bright.

In reality, it is expected that COT$_{warm}$ will tend to increase with CF$_{warm}$, and this is seen in the MODIS-ACAERO retrievals (Eqn. [1] and **Figure S.3**). In contrast, the WRF-CAM5, GEOS and ALADIN simulations show no significant change in COT$_{warm}$ with CF$_{warm}$, so as the cloud field develops to produce greater coverage the clouds are not becoming optically thicker. The range in CF$_{warm}$ covered by the models across the comparison transects included here is smaller than in the observations; a question is whether COT$_{warm}$ would remain largely invariant for CF$_{warm}$ of 0-1.0 in the models. If so, COT$_{warm}$ at lower cloud fractions would be too high in WRF-CAM5, and too low in GEOS and ALADIN. In contrast, in the UM-UKCA simulations COT$_{warm}$ does increase with CF$_{warm}$, at approximately the same rate as in the MODIS-ACAERO observations, but COT$_{warm}$ is systematically too high.

## 5 Summary and conclusions from the comparison

The WRF-CAM5 and GEOS models were used to test for representativeness of the observations of the biomass burning plume along our comparison transects, using aerosol light extinction as the metric (Section 4.1). This approach assumes that the models accurately represent variability in $\sigma_{ep}$, even if there are biases in the mean values of $\sigma_{ep}$. In reality, model variability in aerosol concentrations and properties differ from reality and between models, as seen in the different results when testing for representativeness using both WRF-CAM5 and GEOS. The representativeness of a given set of samples within the models also varies between years, even for the same transect (Meridional2), indicating systematic differences between the models at interannual timescales and/or for different locations. In addition, model invariance in the biomass burning plume aerosol intensive properties (e.g., SSA) made the models not useful for testing the representativeness of sampled aerosol intensive optical properties.





Though there were some exceptions, sampled values of $\sigma_{ep}$ in the 2-5km altitude range across the Diagonal (2016), Meridional1 (2016) and Meridional2 (2017 and 2018) transects were generally within 20% of the approx. month-long period of each years' campaign. This altitude range encompasses the core of the biomass burning plume in most comparison gridboxes and campaign years. The fact that, in the models, the sampled plume concentrations (as measured by $\sigma_{ep}$) were this close to the month-long averages within the campaign years is surprising given that, even with about half the ORACLES flights dedicated to "Routine track" sampling, data was collected for, at most, only 1-2 hours of the full approximately month-long campaign period in each year for a given gridbox and 500m altitude bin (**Figure 2**).

Biases in the plume altitude and concentration were tested through comparisons to observed CO concentration, $\sigma_{ep}$, and BC and OA concentrations across three campaign years covering different months (September 2016, August 2017 and October 2018) and therefore different parts of the African biomass burning season.

Biases in CO for the two models that reported it (WRF-CAM5 and GEOS) suggest under-estimates in emissions or possibly in the efficiency of transport of the biomass burning plume over the SE Atlantic from the burning source regions, since CO is not affected by scavenging processes. An earlier assessment of GEOS representation of the African biomass burning plume indicates that, for that model, transport biases are the more likely cause (Das et al., 2017). In both models, the low bias in CO was larger in October, 2018, which is towards the end of the biomass burning season, than in August, 2017 and September, 2016. Notably, both of these models use GFED emissions.

In the core of the plume, low biases in BC in WRF-CAM5 and GEOS were somewhat smaller than the CO low biases. In other words, the CO:BC ratio was somewhat lower in the models than the observations, particularly in the October 2018 Meridional2 transect, and more so in the GOES model than in WRF-CAM5. The ratio of CO:BC in primary emissions depends on the material being burned and the efficiency of burning (smoldering versus flaming) (e.g. Reid et al., 2005), but this is set within the QFED emissions and so cannot explain these inter-model differences. Thus, they must be due to differences in in-atmosphere chemistry and processing leading to different aerosol scavenging rates. Notably, WRF-CAM5 has more sophisticated aerosol chemistry than GEOS. Greater scavenging losses in the first couple of days after emission, e.g., mostly over the land, in reality than in the models could also be contributing to the lower CO:BC ratio in models than in the observations.

Model biases in BC versus in OA, and how these biases evolve along the aerosol transport pathway, also give some insight to model processes. This is because BC is a refractory, primary aerosol; it is not produced or destroyed in the atmosphere (Bond et al., 2013). OA, on the other hand, is emitted directly as an aerosol as well as being formed in the atmosphere in secondary production from gas phase constituents, and lost in the atmosphere through evaporation back to the gas phase, photo-chemical transformation and/or heterogenous oxidation (e.g. Hallquist et al., 2009; O'Brien and Kroll, 2019).



In most gridboxes above the boundary layer there is a smaller low bias (or, in some locations a greater high bias) in OA than in BC, in the three models that report both parameters (WRF-CAM5, GEOS, and UM-UKCA). The higher OA:BC ratio in 1985 the models, as reflected in their respective relative biases, is more pronounced in 2017 and 2018 and in the GEOS model, and it tends to be lower at the core of the plume than at the plume edges. This could originate from a number of model biases: primary emissions having too high a ratio of OA:BC, too much secondary organic aerosol (SOA) production, and/or insufficient loss of organics with aerosol aging. Again, WRF-CAM5 and GEOS both use the GFED emissions, so should have the same OA:BC ratio in primary emissions. The UM-UKCA model, however uses the FEER inventory which may 1990 have a different OA:BC ratio in emissions from these fires. For OA, in-atmosphere production and losses can be significant. Previous studies have shown that SOA can be produced rapidly – within minutes to hours – after emission (e.g., Jimenez et al., 2009; Bond et al., 2013). OA can also be produced and lost on time-scales of days to >1 week after emission (Capes et al., 2008; Wagstrom et al., 2009; Cubison et al., 2011; Jolleys et al., 2015; Hodzic et al., 2015; Collier et al., 2016; Konovalov et al., 2019; Hodshire et al., 2019; Cappa et al., 2020).

Model-based age estimates indicate that the aerosol we sampled during ORACLES was almost always at least two days old, so our observations don't inform us how the OA:BC ratio evolved shortly after emissions. The processes that drive in-atmosphere OA losses on longer (multi-day) timescales are not implemented in any of the models used here, and this could lead to higher model OA:BC ratios for aged aerosol. However, we cannot rule out too high ratio of OA:BC in emissions and 2000 too much secondary organic aerosol (SOA) production, although the latter is less likely as models tend to show low production of SOA compared to observations (Hodzic et al., 2016).

Biases in $\sigma_{ep}$ stem from the combined biases in aerosol component (e.g., BC and OA) masses and in the mass extinction efficiency. The latter depends in part on aerosol water content, especially above ~40-50% RH. Here, the in-situ observed 2005 values of $\sigma_{ep}$ were made at low (<40%) RH, whereas the HSRL-2 measured and the modeled values were at ambient RH. All four models significantly underestimate in $\sigma_{ep}$, with low biases in $\sigma_{ep}$ greater than for BC or OA. Consistent with this, Shinozuka et al. (2020) calculated a proxy for the mass extinction efficiency, $\sigma_{ep}/(OA+BC)$, from the observations and models, and found it was lower in the models than the observations, including for the 3-6km altitude column. For spherical particles and mid-visible wavelengths, the mass scattering efficiency increases with aerosol diameter between about 100nm 2010 and 450nm diameter, then decreases above about 600nm (e.g., Saide et al., 2020). The SAE provides a proxy for aerosol size; assuming a mono-modal size distribution, typical SAE values indicate an aerosol effective diameter of approx. 380-400nm for ALADIN (SAE of 2.0-2.2), 420nm for the observations and GEOS (SAE of 1.8), 400-420nm for UM-UKCA (SAE of 1.8-2.0), and 700nm for WRF-CAM5 (SAE of 1.0) (Schuster et al., 2006). Because the mass extinction efficiency peaks at about 500nm, aerosol size alone would drive it higher for the observations, GEOS and UM-UKCA models than for 2015 ALADIN – but it should be comparable for the observations and WRF-CAM5 despite the SAE differences (see Figure 10a





of Saide et al., 2020). In this case, a more likely source of the apparent low bias in model mass extinction efficiency are the real indices of refraction used in the models for aerosol components.

The CO, BC, OA, and $\sigma_{ep}$ comparisons all indicate that the models simulated plumes that are too vertically diffuse. Too much diffusion in the models maybe responsible for, in particular, the plume top terminating at lower altitude in the observations than in some of the simulations, often leading to low biases in modeled CO, BC, OA, and $\sigma_{ep}$ in the 2-5km altitude range but significant relative (but small absolute) high biases in the 5-6km range.

In the GEOS, UM-UKCA and ALADIN models it also appears that either the smoke is not lofted sufficiently high over the
continent or that subsidence is too strong in the models, particularly in 2016, but also in 2018 (for GEOS). In the WRF-CAM5 model all biomass burning emissions are injected into the surface-most model layer; this smoke is lifted and mixes in the continental boundary layer, which grows to a depth of typically about 3.5-4.0 km (Labonne et al., 2007) but can reach 4.5-5.5 km (Ryoo et al., 2021). In the UM-UKCA model, emissions added to the boundary layer such that concentrations taper from higher values at the surface to zero at 3 km above the surface. Burning progresses southward through the biomass
burning season with the land surface elevation where burning is occurring shifting from <500m in the Congo-Zaire basin to >1500m in the Namibia-Kalahari dryland. This increase in elevation assists the lofting of the smoke (Ryoo et al., 2021). Notably, the models under-estimate the smoke plume height during the later months of the burning season, when fires are sourced at higher elevations, indicating possible issues with the model representation of boundary layer development over land. It is also possible that lifting of the plume driven by sub-gridscale processes (Freitas et al., 2006) and/or by aerosol
"self-lifting" through absorption and heating (Boers et al., 2010; de Laat et al., 2012) that is not fully accounted for in the models.

The tendency for the models to have too-diffuse a bottom edge of the plume and a plume that is too low in altitude will lead to greater mixing of the aerosol into clouds, and therefore aerosol-cloud interactions. The vertical distance between cloud top
and the biomass burning plume could also affect semi-direct forcing in this region (Adebiyi and Zuidema, 2018). Based on the altitude-dependence to the model bias in aerosol concentrations, the impact of these biases is more pronounced in September 2016 and August 2017 than in October 2018.

While the magnitude of aerosol scattering and absorption over this region is largely controlled by above-cloud AOD, or
vertically-integrated $\sigma_{ep}$, the sign of the direct effect is controlled by the aerosol SSA in the plume and by sub-plume albedo (here, largely controlled by CF$_{warm}$). In the observations, SSA increases with altitude in the plume in September, 2016 and October, 2018 but not in August, 2017. These vertical variations were not captured by any of the models (**Figure 14**). Both WRF-CAM5 and GEOS do, however, have overall higher SSA in August (2018) than in September and October (2016 and 2018), as do the observations.




Co-albedo (1-SSA) differences, weighted by $\sigma_{ep}$, translate directly to differences in absorbed energy. Ambient-RH SSA is lower (and co-albedo higher) than the observed dry SSA in the WRF-CAM5, GEOS and UM-UKCA models; in ALADIN, the SSA is both higher and more variable than observed. These biases vary with altitude, with some of the largest differences in modeled and observed SSA towards the top of the plume. Large SSA biases at altitudes with with very little light

extinction will, however, have little impact on DARE. At altitudes where the plume is largely concentrated (2.5-5km) the, on average the co-albedo in the model is biased high by ~5-10% in UM-UKCA, ~15% in GEOS-5, and 15-20% in WRF-CAM5, relative to the dry observed values. ALADIN co-albedo is biased low by about 10-35% on average. First-order calculations in Section 6.0 demonstrate how these translate into biases in DARE. They will also have impacts for marine low cloud responses to atmospheric absorption above the clouds.


All of the values above are for mid-visible (530 or 550nm) SSA, but of course DARE operates over the full solar spectrum. Spectral SSA is, in turn, directly related to SAE and AAE. Thus, uncertainty in AAE translates into uncertainty in spectral SSA and, in the context of DARE, the amount of sunlight absorbed in the atmosphere. The observed values of $\sigma_{ap}$ are well-constrained at 530nm, but they are very uncertain at shorter and longer wavelengths (470 and 660nm), where the PSAP

measurements have not been as robustly calibrated. For the ORACLES biomass aerosol, the two different Virkkula (2010) calibrations yield AAE values of about 1.2 (wavelength-averaged correction) and 1.5 (wavelength-specific correction), whereas the modeled values average 1.1-1.2, with little difference across the three field campaign years. Notably, the lower values agree well with AAE values measured near Ascension Island during the UK CLARIFY 2017 campaign using a photoacoustic spectrometer to measure absorption (Taylor et al., 2020).


A question is whether this uncertainty in AAE leads to significant uncertainty in DARE. By definition, the AAE parameterization of absorption change versus wavelength is linear in log-space, so for higher AAE the increase in absorption at shorter wavelengths is stronger than the decrease towards longer wavelengths. However, the downwelling solar spectrum in the troposphere peaks at ~450nm and drops off more rapidly at shorter wavelengths than longer wavelengths. The impact

on atmospheric absorption, and therefore DARE, of differing values of AAE results from the convolution of these spectral dependencies of aerosol absorption and downwelling solar radiation.

To quantify this effect, we calculated the 300-750nm integrated atmospheric absorption for aerosol with AAE of 1.0, 1.2 and 1.5, using a fixed 550nm SSA of 0.86, a value of SAE=1.8, and a clear-sky spectral downwelling solar radiation typical of

mid-latitude fall. We find that integrated absorption for AAE=1.5 is only 3% greater than for AAE=1.2 and only 4% greater than for AAE=1.0. The offsetting effects of the spectral dependencies of AAE and solar flux allow DARE to be fairly insensitive to uncertainties in AAE. Therefore the observed model biases in AAE – relative to either of the possible observed



AAE values – will not contribute significantly to biases in modeled DARE, consistent with the findings of de Graaf et al. (2014).


In the ORACLES study region, the sub-plume albedo is a function of CF$_{warm}$, cloud albedo ($\alpha_c$), and the ocean surface albedo. Because of the large difference between cloud albedo and ocean surface albedo, CF$_{warm}$ is a strong controller of sub-plume albedo, with higher CF$_{warm}$ driving more positive DARE. Across our comparison transects, WRF-CAM5 tended to over-estimate CF$_{warm}$, and GEOS and ALADIN to under-estimate CF$_{warm}$, with the UM-UKCA coming closest to

reproducing the observations but still tending to be biased high (**Figure 18**). GEOS and ALADIN in particular also had different gradients in CF$_{warm}$ with latitude (Meridional transects) and longitude (Zonal transect) than was observed; the 2016 Diagonal transect is the only comparison transect where all models largely captured the CF$_{warm}$ gradient. The large difference in COT$_{warm}$ between the observations, WRF-CAM5, GEOS and ALADIN (8-11 for all four) versus in UM-UKCA (24-39) translates to a significant high bias in $\alpha_c$ in the UM-UKCA model, ranging from 40% (2016 Diagonal1 gridbox 3) to 85%

(2016 Meridional gridbox 2) (see Section 6.0 and **Table 3**). This will combine with any high biases in CF$_{warm}$ in the UM-UKCA model to produce too-positive (warming) direct aerosol radiative effects and is sufficient to more than compensate for any small low biases in CF$_{warm}$.

## 6 Impact of aerosol and cloud biases in the Direct Aerosol Radiative Effect

In order to quantify the net effect of model biases on the direct aerosol radiative effect, a first-order DARE estimate is

calculated using the gridbox-mean aerosol and cloud properties for five of the comparison gridboxes. The 2016 Diagonal gridbox 3 was selected for being closer to the center of the plume while (in contrast to gridboxes 1 and 2) having more robust sampling (**Figures 1** and **2**); 2016 Meridional1 gridbox 2 and 2017 and 2018 Meridional2 gridbox 5 for being located closer to the center of the plume meridionally; and 2017 Meridional2 gridbox 2 in order to have one gridbox with lower cloud fraction (57%), since the other four selected gridboxes all have average CF$_{warm}$ of >75%.


Following Haywood and Shine (1995), DARE at the top of the atmosphere can be estimated as

$$DARE \approx -D \cdot S_o \cdot T_{at}^2 \cdot (1 - A_c) \cdot SSA \cdot \bar{\beta} \cdot AOD \cdot \left((1 - R_s)^2 - \frac{2\,R_s}{\bar{\beta}}\left(\frac{1}{SSA} - 1\right)\right) \tag{2}$$

where $D$ is the daylight fraction of the day; $S_o$ is the solar constant; $T_{at}$ is the atmospheric transmissivity (absent aerosol); $A_c$ is cloud fraction; SSA is the single scatter albedo; $R_s$ is the surface reflectance; $\bar{\beta}$ is the spectrally-weighted aerosol

hemispheric backscatter fraction; and $AOD$ is the aerosol optical depth. This formulation assumes zero forcing in the presence of clouds (since for $A_c=1$, DARE=0). The goal here is to calculate the forcing for an aerosol plume that, when clouds are present, is fully above the cloud layer and therefore has non-zero forcing. Equation [2] is therefore modified to:



$$DARE \approx -D \cdot S_o \cdot T_{at}^2 \cdot SSA \cdot \bar{\beta} \cdot AOD \cdot \left( (1 - \alpha_s)^2 - \frac{2\,\alpha_s}{\bar{\beta}} \left( \frac{1}{SSA} - 1 \right) \right) \qquad [3]$$

where $\alpha_s$ is the "scene" albedo below the aerosol plume and AOD is the above-cloud AOD. In this formulation, the impact
of clouds on DARE is accounted for through their effect on $\alpha_s$.

DARE is non-linear with $\alpha_s$ (Eqn. [3] and Cochrane et al., 2021), so here DARE is calculated separately for clear
(*DARE$_{clear}$*) and cloudy (*DARE$_{cloudy}$*) skies. The gridbox-average DARE (*DARE$_{avg}$*) is the sum of the two, weighted by their
average fractional contributions in each gridbox:

$$DARE_{avg} = DARE_{clear}(1 - CF_{warm}) + DARE_{cloudy}CF_{warm}. \qquad [4]$$

This assumes that the aerosol is not systematically different over clear skies than over cloudy skies, as demonstrated for this
region by Kacenelenbogen et al. (2019). The observed values of CF$_{warm}$ used in Eqn. [4] are the gridbox means from the
SEVIRI-LaRC retrievals.

For clear skies, $\alpha_s$ in Eqn. [3] is set to 0.07 (approximated from Li et al., 2006 for ocean reflectivity), and for cloudy skies,
$\alpha_s$ is the gridbox-average cloud albedo. The two-stream approximation of Feingold et al. (2017) is used to calculate the
visible cloud albedo $\alpha_c$:

$$\alpha_c = \frac{\frac{(1-g)}{\cos \theta_o} \tau_c}{2 + \frac{(1-g)}{\cos \theta_o} \tau_c}, \qquad [5]$$

where the asymmetry factor, $g$, is set to 0.85 (Bohren, 1980), the solar zenith angle, $\theta_o$, is fixed at 30°, and cloud optical
thickness, $\tau_c$, is set to the gridbox log-mean value of COT$_{warm}$ from the MODIS-ACAERO retrievals.

In Eqn. [3], $D$ is fixed at 0.5 (which is correct to within 0.02 for this latitude range in all three months), $S_o$ is set to 1361 Wm$^{-2}$ (Kratz et al., 2020), and $T_{at}$ is set to 0.79 based on Figure 1 of Wild et al. (2019). The value $\bar{\beta}$ is calculated using Eqn. 11b
of Reid et al. (1999), which parameterizes $\bar{\beta}$ as a function of $\sigma_{ep}$ at 550nm and the extinction Ångström exponent (EAE)
across 437-669nm (close to our wavelength-span of 470-660nm) for biomass burning smoke. For the gridboxes included in
this analysis, modeled $\bar{\beta}$ values varied from a low of 0.094 (in WRF-CAM5) to a high of 0.159 (in ALADIN), with observed
values in the range 0.11-0.13 (**Table 3**). EAE, used in deriving $\bar{\beta}$, is calculated as:

$$EAE = SSA \cdot SAE + (1 - SSA) \cdot AAE . \qquad [6]$$

The values of SSA, AAE and SAE used in Eqns. [3] and [6] are extinction-weighted column values and AOD the integral of
$\sigma_{ep}$, all calculated across 1.5-5.5 km altitude, since this altitude range captures the vast majority of the above-cloud smoke
plume in both the observations and the models.





AAE was not reported for the ALADIN model, so the observed value is used in the calculation of DARE (**Table 3**). In addition, in 2018 there were problems with the measurement of SAE, so the observed value in that year is set to 1.8, since 2145 SAE in 2016 and 2017 was typically 1.7-1.9 at plume altitudes across most comparison gridboxes. Aerosol properties for the UM-UKCA model are the ambient-RH values.

Equations [3] and [4] provide a valuable tool to represent the functional dependency of DARE on aerosol and cloud properties, as well as surface albedo. However, the resulting values (**Table 3**) are an approximation that does not fully 2150 account for all of the factors that influence DARE. For example, a fixed solar zenith angle (30°) is used in calculating cloud albedo, the aerosol backscatter (rather than up-scatter) fraction is used in the calculation, and this formulation doesn't account for the effects of sun angle on atmospheric gaseous transmission ($T_a$) and on aerosol scattering; spectral variations in aerosol and radiative properties are not included either. The amount of sunlight that interacts with the aerosol at a given altitude also depends on extinction by aerosol at higher altitudes, and this is not accounted for in using a fixed atmospheric 2155 transmission factor. In addition, the calculation uses month-long gridbox averages for aerosol and cloud properties. DARE does not scale linearly with SSA or the sub-plume albedo, and therefore with $CF_{warm}$ and $COT_{warm}$, so the "mean" DARE values presented here will differ from a mean of instantaneous DARE values.

To explore the limitations of Eqn. [3], we perform full radiative transfer calculations for spectrally-resolved upward and 2160 downward broadband fluxes (100nm bandwidth in the visible, and 500nm beyond) for cloudy and clear skies, with and without aerosol, using libRadTran (Mayer and Kylling, 2005). From these fluxes, we calculate the instantaneous DARE in cloudy and clear skies. Finally, $DARE_{avg}$ is calculated from the cloud fraction weighted DARE of both cloudy and clear skies, as in Eqn. [4]. To compare with the parameterized $DARE_{avg}$, the instantaneous $DARE_{avg}$ are integrated over 24 hours to obtain the diurnally-averaged DARE. Our simulations use a mid-latitude winter gas profile and correlated-k 2165 parameterization from Kato et al. (1999) for spectrally resolved results prior to spectral integration. The dark ocean in the simulation is treated as a Lambertian surface with prescribed wavelength-dependent albedo. A slab aerosol layer is assumed for 2-5 km altitude, and the cloud layer is located at 0.7-1 km. Spectral AOD and SSA are calculated using the EAE and AAE in **Table 3**.

The comparison of parameterized $DARE_{avg}$ with the full radiative transfer calculations are shown in Supplementary **Figure S.9**. We find a high correlation coefficient ($R^2=0.95$), with relatively few outliers, which are mostly confined to the $DARE_{avg}$ estimates for 2017 Meridional2 gridbox2. We conclude that the parameterized $DARE_{avg}$ estimates in **Table 3** are useful for providing a first-order estimate of how biases in key aerosol and cloud properties translate to biases in DARE in this region, and we proceed by using the parameterized $DARE_{avg}$ expressions to assess the contribution of each of the variables in **Table** 2175 **3** to a bias in derived $DARE_{avg}$. These contributions are indicated in square brackets following the mean value of that variable; they are the ratio of $DARE_{avg}$ calculated using that model's value for the given variable only and the observed





**Table 3.** TOA DARE$_{avg}$ (in W m$^{-2}$) calculated using Eqn. [4], and the gridbox-mean observed and modeled aerosol and cloud properties used in the calculation, for select comparison gridboxes. Also given in square brackets after each modeled variable is the ratio of DARE$_{avg}$ calculated using that model's value for the given variable only and the observed values for all other variables, to DARE$_{avg}$ calculated using the observed variables. The values of $\bar{\beta}$, $\alpha_c$ and $\alpha_s$ are shown in italics because they are derived from the observed and modeled parameters (see text). In the table below, $\alpha_s = 0.07(1 - CF_{warm}) + \alpha_c CF_{warm}$ is given as a metric for the combined effect of CF$_{warm}$ and COT$_{warm}$ on DARE. As noted in the text, DARE is actually calculated separately for clear and cloudy skies using Eqn. [3], then DARE$_{avg}$ calculated using Eqn. [4].

| | observed | WRF-CAM5 | GEOS | UM-UKCA | ALADIN |
|---|---|---|---|---|---|
| | | *2016 Diagonal, gridbox 3* | | | |
| AOD | 0.360 | 0.254 [*0.71*] | 0.226 [*0.63*] | 0.181 [*0.50*] | 0.276 [*0.77*] |
| SSA | 0.855 | 0.841 [*1.14*] | 0.851 [*1.04*] | 0.842 [*1.13*] | 0.892 [*0.64*] |
| SAE | 1.77 | 1.05 [*1.07*] | 1.75 [*1.00*] | 1.73 [*1.01*] | 2.08 [*0.90*] |
| AAE | 1.48 | 1.12 [*1.01*] | 1.13 [*1.01*] | 1.38 [*1.00*] | {1.48} [*N/A*] [1] |
| CF$_{warm}$ | 0.88 | 0.96 [*1.16*] | 0.52 [*0.30*] | 0.85 [*0.94*] | 0.64 [*0.53*] |
| COT$_{warm}$ | 11.5 | 7.6 [*0.63*] | 6.2 [*0.44*] | 24.4 [*1.61*] | 9.9 [*0.86*] |
| $\bar{\beta}$ | *0.120* | *0.094* | *0.112* | *0.114* | *0.158* |
| $\alpha_c$ | *0.499* | *0.397* | *0.349* | *0.678* | *0.461* |
| $\alpha_s$ | *0.448* | *0.384* | *0.215* | *0.587* | *0.320* |
| DARE$_{avg}$: | **14.8** | **9.9** | **0.3** | **12.7** | **-0.1** |
| model/observed DARE: | | **0.67** | **0.02** | **0.86** | **-0.01** |
| | | *2016 Meridional1, gridbox 2* | | | |
| AOD | 0.419 | 0.257 [*0.62*] | 0.236 [*0.56*] | 0.259 [*0.62*] | 0.375 [*0.90*] |
| SSA | 0.867 | 0.833 [*1.73*] | 0.868 [*0.98*] | 0.854 [*1.28*] | 0.885 [*0.62*] |
| SAE | 1.89 | 1.20 [*1.42*] | 1.65 [*1.24*] | 1.90 [*0.99*] | 2.05 [*0.75*] |
| AAE | 1.57 | 1.15 [*1.08*] | 1.14 [*1.08*] | 1.47 [*1.02*] | {1.57} [*N/A*] [1] |
| CF$_{warm}$ | 0.76 | 0.90 [*1.61*] | 0.52 [*-0.04*] | 0.92 [*1.70*] | 0.57 [*0.17*] |
| COT$_{warm}$ | 8.3 | 8.2 [*0.99*] | 5.1 [*0.01*] | 38.7 [*3.77*] | 8.6 [*1.10*] |
| $\bar{\beta}$ | *0.133* | *0.096* | *0.108* | *0.137* | *0.158* |
| $\alpha_c$ | *0.417* | *0.415* | *0.306* | *0.770* | *0.427* |
| $\alpha_s$ | *0.334* | *0.381* | *0.193* | *0.714* | *0.274* |
| DARE$_{avg}$: | **6.2** | **10.4** | **-1.1** | **21.4** | **-2.4** |
| model/observed DARE: | | **1.68** | **-0.18** | **3.44** | **-0.39** |
| | | *2017 Meridional2, gridbox 2* | | | |
| AOD | 0.292 | 0.450 [*1.54*] | 0.352 [*1.21*] | 0.258 [*0.88*] | 0.485 [*1.66*] |
| SSA | 0.861 | 0.835 [*1.91*] | 0.866 [*0.82*] | 0.854 [*1.25*] | 0.888 [*0.05*] |
| SAE | 1.70 | 1.02 [*1.56*] | 1.67 [*1.05*] | 1.72 [*0.96*] | 2.06 [*-0.09*] |
| AAE | 1.44 | 1.1 [*1.10*] | 1.14 [*1.09*] | 1.34 [*1.03*] | {1.44} [*N/A*] [1] |
| CF$_{warm}$ | 0.57 | 0.74 [*2.46*] | 0.46 [*0.07*] | 0.80 [*2.91*] | 0.33 [*-1.06*] |
| COT$_{warm}$ | 8.0 | 8.5 [*1.20*] | 6.9 [*0.53*] | 26.8 [*4.52*] | 7.0 [*0.57*] |
| $\bar{\beta}$ | *0.114* | *0.094* | *0.109* | *0.113* | *0.160* |
| $\alpha_c$ | *0.410* | *0.425* | *0.374* | *0.699* | *0.377* |
| $\alpha_s$ | *0.264* | *0.333* | *0.210* | *0.571* | *0.170* |
| DARE$_{avg}$: | **2.1** | **14.0** | **-0.7** | **15.6** | **-12.9** |





| model/observed DARE: | | **6.58** | **-0.35** | **7.35** | **-6.08** |
|---|---|---|---|---|---|
| *2017 Meridional2, gridbox 5* | | | | | |
| **AOD** | 0.591 | 0.370 [*0.62*] | 0.389 [*0.65*] | 0.252 [*0.42*] | 0.562 [*0.94*] |
| **SSA** | 0.868 | 0.830 [*1.50*] | 0.837 [*1.40*] | 0.852 [*1.20*] | 0.874 [*0.91*] |
| **SAE** | 1.80 | 1.10 [*1.14*] | 1.77 [*1.00*] | 1.77 [*1.00*] | 2.14 [*0.74*] |
| **AAE** | 1.55 | 1.11 [*1.02*] | 1.14 [*1.02*] | 1.30 [*1.01*] | {1.55} [*N/A*][1] |
| **CF$_{warm}$** | 0.81 | 0.87 [*1.15*] | 0.67 [*0.62*] | 0.92 [*1.28*] | 0.53 [*0.24*] |
| **COT$_{warm}$** | 10.4 | 7.8 [*0.65*] | 5.4 [*0.22*] | 34.7 [*2.22*] | 9.98 [*0.94*] |
| $\bar{\beta}$ | *0.122* | *0.095* | *0.114* | *0.115* | *0.171* |
| $\alpha_c$ | *0.473* | *0.403* | *0.319* | *0.750* | *0.464* |
| $\alpha_s$ | *0.397* | *0.360* | *0.237* | *0.696* | *0.279* |
| **DARE$_{avg}$:** | **16.0** | **14.0** | **3.4** | **20.7** | **-3.1** |
| model/observed DARE: | | **0.88** | **0.21** | **1.30** | **-0.22** |
| *2018 Meridional2, gridbox 5* | | | | | |
| **AOD** | 0.248 | 0.165 [*0.67*] | 0.261 [*1.06*] | --- | --- |
| **SSA** | 0.890 | 0.851 [*2.04*] | 0.877 [*1.35*] | --- | --- |
| **SAE** | 1.8[3] | 1.22 [*1.36*] | 1.68 [*1.13*] | --- | --- |
| **AAE** | 1.45 | 1.18 [*1.05*] | 1.13 [*1.05*] | --- | --- |
| **CF$_{warm}$** | 0.77 | 0.95 [*1.87*] | 0.31 [*-1.21*] | --- | --- |
| **COT$_{warm}$** | 9.0 | 8.5 [*0.88*] | 2.7 [*-1.42*] | --- | --- |
| $\bar{\beta}$ | 0.123 | 0.096 | 0.111 | --- | --- |
| $\alpha_c$ | 0.438 | 0.424 | 0.190 | --- | --- |
| $\alpha_s$ | 0.353 | 0.406 | 0.107 | --- | --- |
| **DARE$_{avg}$:** | **3.1** | **6.4** | **-5.7** | --- | --- |
| model/observed DARE: | | **2.09** | **-1.85** | --- | --- |


values for all other variables, to DARE$_{avg}$ calculated using the observed variables. For example, for the 2016 Diagonal gridbox 3, the WRF-CAM5 low bias in SSA alone drives a 14% high bias in DARE$_{avg}$.

Notably, DARE$_{avg}$ is positive (>0) across all five gridboxes when calculated using the observed aerosol and cloud properties and the properties from the WRF-CAM5 and UM-UKCA models. In contrast, it is negative in all five gridboxes using the properties simulated with ALADIN (**Table 3**) and negative for three of the five gridboxes for GEOS. The modeled ambient-RH 1.5-5.5 km AOD is lower than the observed values at low RH for almost all gridboxes and models; even in only those gridboxes where observed AOD is higher (see Meridional2 in both 2017 and 2018 in **Table 3**), DARE$_{avg}$ calculated from the

model properties can be either larger and more positive or smaller and of the opposite sign to DARE$_{avg}$ calculated from the observed properties. This is because biases in different variables often counteract each other.

This can be seen in the calculations for gridbox 2 in the 2017 Meridional2 transect. This example stands out in that the modeled AOD is larger than observed in three of the four models (WRF-CAM5, GEOS and ALADIN). CF$_{warm}$ in this

gridbox is also lower than in the other four gridboxes (0.57 versus >0.75). In the WRF-CAM5 model this high AOD bias





combines with, in particular, a high bias in modeled $CF_{warm}$, and a low bias in SSA to produce values of $DARE_{avg}$ that is a factor of 6.6 larger than when calculated using the observed properties. In contrast, in ALADIN an AOD high bias of a similar magnitude combines with a significant low bias in $CF_{warm}$, and a high bias in SSA to produce a $DARE_{avg}$ that is 6.1 times larger than that using observed properties – but of the opposite sign.


Across the five gridboxes, biases in $\sigma_{ep}$ (AOD) and $CF_{warm}$ and $COT_{warm}$ (through their role in determining $\alpha_s$) alternately make the largest contributions to biases in $DARE_{avg}$ in many of the models/gridboxes. Biases in SSA are usually the source of somewhat smaller $DARE_{avg}$ biases, but it is a still significant contributor across most models and gridboxes. Low biases in SAE in the WRF-CAM5 and GEOS-5 models and a high bias in SAE in the ALADIN model also can drive biases on the
order of 30-40% through its role in determining $\bar{\beta}$ (see Eqn. [3]).

$DARE_{avg}$ derived from the WRF-CAM5 properties vary from being within about 30% of $DARE_{avg}$ calculated using the observed properties in two of the five gridboxes to having significant high biases in the other three (factors of 1.7, 6.6 and 2.1). In these latter three cases, the consistent low bias in SSA and high bias in $CF_{warm}$ more than offsets a low bias in AOD
in two of the gridboxes and adds to the high bias in AOD in the third. This is not the case in 2016 Diagonal gridbox 3, where SSA and $CF_{warm}$ are still biased low and high, respectively, but combined with a low bias in $COT_{warm}$, the resulting $DARE_{avg}$ is much close to $DARE_{avg}$ from the observed properties. The excellent agreement in 2017 Meridional 2 gridbox 5 in DARE using the observed and WRF-CAM5 properties also results from compensating biases: the simulated AOD and $COT_{warm}$ bias $DARE_{avg}$ low and simulated SSA and (to a lesser degree) SAE and $CF_{warm}$ bias $DARE_{avg}$ high, nearly exactly offsetting each
other. This shows how qualitatively consistent biases in a given model's representation of aerosol and cloud properties key to DARE can combine to produce a large range of biases in modeled DARE.

In GEOS, the extinction-weighted SSA in the plume introduces small (<20%) low biases in $DARE_{avg}$ in three of the gridboxes and 35-40% low biases in the other two. More significant for this model is larger low biases in AOD in three of
the five gridboxes and systematic low biases in $CF_{warm}$, and $COT_{warm}$. These result in values of $DARE_{avg}$ that are too small and, for three of the five cases, produce negative rather than positive values. As in WRF-CAM5, biases in different variables can offset each other. For example, correcting for AOD alone would produce even larger negative values of forcing for three of the five gridboxes, increasing the difference between $DARE_{avg}$ from the modeled and observed properties.

In the UM-UKCA model, a high bias in $CF_{warm}$ and a large high bias in $COT_{warm}$ make the sub-plume scene albedo on the order of a factor of two too high, driving large high biases in $DARE_{avg}$. This, combined with a small (~0.01-0.02) low bias in SSA, more than compensates for the low bias in AOD to produce values of $DARE_{avg}$ that are too large in three of the four gridboxes tested, and significantly so (by factors of 3.4 and 7.4) in two of the gridboxes. Correction to just the aerosol fields




would produce universally too-high DARE$_{avg}$; correction to just the cloud fields, conversely, would produce universally too-
low DARE$_{avg}$.

The ALADIN model consistently simulates too-small CF$_{warm}$ and too-high SSA. These are sufficient to produce a negative direct aerosol radiative effect across all five gridboxes, in contrast with the positive DARE$_{avg}$ calculated from the observed values. In 2017 Meridional2 gridbox 2 the CF$_{warm}$ bias alone is sufficient to produce negative DARE. Small low biases in
AOD and COT$_{warm}$ and a high bias in SAE also combine to produce values of DARE$_{avg}$ that are too small (AOD and SAE) and too negative (COT$_{warm}$).

These findings support that of earlier studies (e.g. Chand et al., 2009; Stier et al. 2013; Zuidema et al., 2016) that emphasize the importance of accurately simulating cloud fraction in order to get accurate estimates of direct forcing by biomass burning
aerosol over the SE Atlantic. They also highlight the importance of quantifying the relative roles of the aerosol and cloud properties that control the direct aerosol radiative effect, since the magnitude, sign and source(s) of DARE biases can vary with the aerosol and cloud properties themselves. The limited analysis presented here makes it clear that models can have compensating biases, such that model improvements that make one key parameter more accurate could actually lead to less accurate simulated values of DARE. This first-order analysis provides a framework for a future study that employs full
radiative transfer calculations that use, for example, the full 2-D profiles of observed and modeled cloud properties and account for diurnal effects, and that accounts for uncertainties and variability in observed and modeled fields.

**Data availability**

The P-3 and ER-2 observational data are available through:
https://espoarchive.nasa.gov/archive/browse/oracles/id8/ER-2 (ER-2 2016 data)
https://espoarchive.nasa.gov/archive/browse/oracles/id14 (2017 P-3 data)
https://espoarchive.nasa.gov/archive/browse/oracles/id22 (2018 P-3 data)
The aggregated model and observational products are available at:
https://espo.nasa.gov/sites/default/files/box_P3ER2Models_2016mmdd_R8.nc

**Author contributions**

SPB, RF, AD, SF, SGH, JRP, operated instruments during the ORACLES intensive observation periods. PES, CH, GAF, HG, MM, PN, GRC, AdS, delivered model products. KM and DP provided and assisted with analysis of satellite cloud



products for the ORACLES intensive observation periods. PES, SJD, JR, RW and PZ formulated the model-observation comparison. SD processed all observational data and applied statistical techniques. SJD wrote most of the first draft. JR, IC and LG provided the full radiative transfer calculations for testing the parameterized DARE and wrote the associated text. SD, PES, PZ, GAF, HG, MM, KM, DM, SPB, RF, KP, RW and JR edited the manuscript. JR, RW and PZ led the efforts to acquire funding for the ORACLES mission.

**Competing Interests**

The authors declare that they have no conflict of interest.

**Acknowledgements**

All in-situ aerosol data were collected as part of the ORACLES project, which is a NASA Earth Venture Suborbital 2 investigation funded by NASA's Earth Science Division and managed through the Earth System Science Pathfinder Program Office. The ORACLES team gratefully acknowledges the work by the NASA Ames Earth Science Project Office (ESPO), led by Dr. Bernadette Luna and Mr. Dan Chirica. The team is equally grateful for the tireless contributions by the NASA Wallops and NASA Johnson P-3 and ER-2 pilot and flight crews, as well as air traffic control at Walvis Bay airport (Namibia) and the airport in São Tomé. Local authorities in Namibia and São Tomé played important roles as well, for which the project would like to express their gratitude.

We would like to thank the many groups and individuals involved with producing the satellite-retrieved cloud properties used in this study. For the "MODIS-Standard" cloud retrievals, the Terra and Aqua MODIS Level 3 cloud data were acquired from the Atmosphere Archive & Distribution System (LAADS) Distributed Active Archive Center (DAAC), located in the Goddard Space Flight Center in Greenbelt, Maryland (https://ladsweb.nascom.nasa.gov/).

For the SEVIRI-LaRC cloud retrievals, we acknowledge the SatCORPS team at NASA-LaRC with the webpage https://satcorps.larc.nasa.gov/ and include the following papers for a description of the retrievals: Minnis et al. (2008; 2011a; 2011b) and Painemal et al. (2015).

The MODIS-ACAERO retrievals are available from the author.

The WRF-CAM5 simulations were funded with Department of Energy Grant number DOE ASR DE-SC0018272.
This publication is partially funded by the Cooperative Institute for Climate, Ocean and Ecosystem Studies (CICOES) under NOAA Cooperative Agreement NA15OAR4320063, Contribution No. 2020-1126.



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
