# Peer review of "Modeled and observed properties related to the direct aerosol radiative effect of biomass burning aerosol over the Southeast Atlantic"

_Atmospheric Chemistry and Physics, 2021_

## Author Comment (AC1)

**Replies to reviews of acp-2021-333-RC1, 2021**

*Reviewer comments are shown in black, non-italicized text; replies are in blue italicized text.*

**Comment on acp-2021-333**

Anonymous Referee #1

Referee comment on "Modeled and observed properties related to the direct aerosol radiative effect of biomass burning aerosol over the Southeast Atlantic" by Sarah J. Doherty et al., Atmos. Chem. Phys. Discuss., https://doi.org/10.5194/acp-2021-333-RC1, 2021

Review for "Modeled and observed properties related to the direct aerosol radiative effect of biomass burning aerosol over the Southeast Atlantic" by Doherty et al.

This study presents a thorough comparison between a number of modelling frameworks and in situ observations made during the ORACLES field campaign that took place over the southeast Atlantic Ocean during three month-long periods in three consecutive years. The study focuses on parameters and quantities that are used in quantifying the direct radiative effect from biomass burning aerosol. A first order calculation of the direct effect using observations and comparisons to the models is presented. The study helps identify key failings in the ability of the models to reproduce the observations, which will be very useful for focusing future studies.

Although the manuscript is considerably long I believe the in-depth presentation of methods and the authors' treatment of uncertainties is on the whole necessary. However, I believe some figures and passages of text can be improved to improve the flow of the study and highlight the key messages being presented throughout. I therefore recommend this manuscript is published in ACP once the, largely minor, comments below are addressed.

*We thank the reviewers for the overall positive review, and for the helpful guidance on areas where it could be improved.*

Main comments:

My main concern is the length of the manuscript. On reading it I often found that the key messages in each section or paragraph were not as clear as they could be. For shorter manuscripts this would be fine, but due to the length of this manuscript I strongly suggest the authors rethink some aspects.

*We have tried to edit the paper in places for better clarity and conveyance of key messages. In particular, the discussion of the extinction comparisons is now broken into sub-sections, and the discussions of the SSA and SAE biases (hopefully!) improved.*

The figures are very large and I found myself endlessly scrolling through them. I suggest replotting the figures to make them smaller and more compact. For instance, try and combine Figures 7-9, same for 10-13, etc.

*Figures 7-9 and 10-13 have been combined to show only the mean biases, with results for all models in one plot. The figures showing individual gridbox mean biases have been moved to the Supplemental document.*

The second aspect that can be improved is the construction of the paragraphs. The manuscript occasionally contains sections consisting of short paragraphs (for example see lines 1560 to 1580, 1600 to 1625) which break the flow of the manuscript and makes it difficult to identify the key messages. To make the manuscript more readable I suggest the authors go through the manuscript and make sure the key messages for each section are clearly delivered. Some paragraphs can be reduced in length with just the key result put forward – the addition of exact values and consideration of individual grid boxes sometimes made it difficult reading.

*The referenced sections, and others, have been edited for improved clarity.*

I don't feel that section 4.1 brings much to the manuscript. I believe the methodology is somewhat flawed and the outcomes are not used in the rest of the manuscript. Using the models to provide the test of representativeness is entirely dependent on the ability of the models to reproduce the observations – which as shown later in the manuscript isn't great. This is further illustrated by the lack of consistency between the two models used. At the end of the section there is no discussion on what the results actually tell us and how they influence the rest of the manuscript. Even in the summary it is difficult to understand what the outcomes of the analysis are. Does this tell us anything that Shinozuka et al. (2020) does not? Does the analysis mean that the proceeding comparison is not representative of the climatology? I suggest either removing the section or making the outcomes clearer.

*It is acknowledged in the text that "Comparison of the two allows for testing the representativeness of the observations for assessing monthly averages, assuming the model realistically captures aerosol variability." Note that to test for representativeness the models need to get the relative variability in the aerosol correct – but not necessarily it's mean properties. It turns out the models do okay in representing variability, at least for the 2016 transects; and one of each of the models does well for the other two transects. We have now added the following analysis and text to the manuscript in Section 2.2:*

*"To test this, we calculated the fractional variability in $\sigma_{ep}$ in the 2-5km altitude range within the comparison gridboxes in the WRF-CAM5 and GEOS models and in the observations. We found that the three are similar for the 2016 Diagonal and Meridional1 transect gridboxes, with the models alternately having lower and higher fractional variability than was observed. Along the 2017 Meridional2 transect, WRF-CAM5 and the observed variability were similar but the GEOS variability was 10-30% higher. Along the 2018 Meridional2 transect the converse was true: the relative variability in $\sigma_{ep}$ was similar for GEOS and the observations, but was about 20% lower for WRF-CAM5."*

*Text in Section 4.1 is now edited to incorporate this new analysis.*

The DARE section is the highlight of the paper. In its current form it feels like a second paper that was appended onto the manuscript. The authors may wish to consider moving the summary section to the end and integrate the DARE section into the manuscript.

*We're glad the reviewer liked this analysis. The DARE section has been moved to before the final Discussion and Conclusions section. The latter now includes a summarizing paragraph about how model biases contribute to biases in DARE, so it covers the whole paper.*

Minor comments:

Line 61. This one sentence hides a lot of the importance and uncertainties and of ARI and ACI. Maybe expand to give a better account of why understanding ARI and ACI is important?

*As the issue of aerosol climate forcing is so commonly known and written about in the scientific community we were trying not to include a lot of background information that is already well known by the community. To give some context of the magnitude of its importance we now precede this with the sentence:*
*"Climate forcing by both direct aerosol-radiation interactions and aerosol-cloud interactions offsets about a third of greenhouse gas forcing and also contributes the largest uncertainty to total anthropogenic forcing (Forster et. al., 2021)."*

Line 243. How sensitive is the weighting function to the chosen value for the standard deviation?

*A sentence has been added: "A much larger standard deviation would have weighted values more than 30sec from the time of interest too heavily (e.g. by 17% for a value of 16), and a much smaller standard deviation would have produced a weighting function that approached zero in less than 30sec."*

Line 250. Please provide a characteristic size for the accumulation mode.

*Text now reads: "Above the boundary layer, the aerosol during ORACLES was dominated by accumulation mode biomass burning smoke with a volumetric mean diameter of <0.4μm (e.g. see Figure 8 of Shinozuka et al., 2020), so …"*

Line 283. Please clarify what the 'original PSAP instrument' is referring to.

*This has been edited to read: "Early versions of the PSAP instrument …"*

Line 305. It may be useful to include a sentence that sums up what satellite product you end up assigning to each cloud property variable.

*Good idea. The following has been added to the end of this section: "CF$_{warm}$ and COT$_{warm}$ are derived for several retrieval products and are compared to each other as well as to the observations. As described later (Section 4.4) the SEVIRI-LaRC values of CF$_{warm}$ (Section 3.2.3) and the MODIS-ACAERO values of COT$_{warm}$ (Section 3.2.2) are used as the benchmark for the comparison to the modelled values."*

Line 388. Can you include a concluding sentence that answers the question at the beginning of this paragraph?

*The following sentence has been added: "As such, the COT$_{warm}$ values from MODIS appear to represent very well the daytime average, within the context of their role in determining aerosol direct radiative effects."*

Line 485. The figure shows ratios and the text discusses percentages. Please change one of them to make it consistent.

*The text has been revised to discuss ratios.*

Line 508. Doesn't Figure 3f show a bias up to +200% at one level? Or are you talking about the column mean?

*Apologies; this was referring to the 2-3km and 3-4km altitude bins but that wasn't specified. The next sentence notes the large high biases above 4km. The text now reads: "Despite this, in the heart of the plume (2-4km altitude), sampled values of $\sigma_{ep}$ were generally with 0.9-1.2 of the month-long climatology in the both models (Figure 3e,f). Both models also indicate that smoke concentrations sampled by the P-3 at higher altitudes (4-6km) are much higher than was typical."*

Line 700. Is this consistent with Shinozuka's conclusions for representativeness?

*Shinozuka et al. barely mentioned the representativeness of the sampled SSA (or other aerosol intensive properties). Text has been added noting that the results here for extinction representativeness in 2016 are consistent with those of Shinozuka et al. (2020).*

Line 940. Do the models also show weak wet deposition?

*Determining this would require that metrics for wet deposition were examined for the models and we unfortunately didn't think in advance to save this information. We can only note that the models, like the observations, show some mid- and high-level (cirrus) clouds along the eastern end of the ORACLES study range, near the west African coast, and otherwise simulate varying amounts of low marine clouds. As such there is likely very little to no wet deposition from clouds above the marine boundary layer. The exception is the very southern end of the study region, which was sometimes affected by the very northern end of frontal systems. However, this would only have affected perhaps the two southern-most gridboxes of the 2016 Diagonal transect. As all of this is very hand-waving and inferential we have not added anything on this to the text.*

Line 956-958. Are you discussing the characteristics of the observations or comparing against the models?

*We were referring to what the observations show. To clarify this now reads: "On average, the observed core of the plume …"*

Line 958' "a broader vertical extent towards the core of the plume". This doesn't make sense to me.

*We can see how this might be confusing, as "core" can refer to the center of the plume vertically or horizontally. This text has now been edited to read: "a broader vertical extent towards the geographic center of the plume".*

Line 963. There doesn't seem to be much consistency in the 'over-estimation above and below the plume centre' for WRFCAM in Figure 7 or in table 1.

*This text has been edited to now read: "This overly vertically diffuse plume in WRF-CAM5 is consistent with underestimates in BC concentrations in the core of the plume (3-4 km altitude in 2016 and 2018; 3.5-5km in 2017) (Figures 12 and 13), over-estimates above this, and smaller biases for the 2-3km altitude bin. The pattern of bias is similar for GEOS, except that it does a better job of reproducing BC concentrations aloft at the southern end of the 2016 Meridional1 transect and has greater high biases at the bottom of the plume (2-3km altitude bin)."*

Line 965. I'm surprised you don't point out the substantial and largely consistent overestimation at lower altitudes for GEOS.

*The focus of this study is the biases in the plume, which is in the free troposphere so the text was addressing that. However, it's correct that GEOS does indeed have large high biases in BC and OA in the boundary layer. As this was pointed out for UM-UKCA only, the omission of noting this for GEOS does stand out. The last sentence in this paragraph now reads: "Both GEOS and UM-UKCA over-estimates the amount of BC (i.e. biomass smoke) that mixes into the marine boundary layer, with consequences for derived forcing through aerosol-cloud interactions."*

Line 1136. Spelling mistake: humidification

*Thank you. Corrected.*

Line 1144. Do you have a sense of how sensitive your results would be to gamma?

*The following sentence has been added: "If γ is actually lower than derived here – e.g., closer to a value of ~0.3, as measured in the year 2000 SAFARI campaign in southwest Africa (Titos et al., 2016) – the f(RH) values applied here would be about 30% too large (~1.5 vs. ~1.2 for γ of 0.65 vs. 0.3, for an ambient RH of 60%)."*

Line 1152. Please can you include the calculated ranges?

This paragraph now reads:

*"Shinozuka et al., (2020) estimated that, for September 2016, f(RH) was less than 1.2 for 90% of the free troposphere aerosol measurements across the campaign. Estimates for the comparison gridboxes included here are consistent with this (Figure S.14), but also show that f(RH) for the 2016 Diagonal gridboxes 3-6 in the 4-5.5 km altitude range were often higher, with means of 1.30±0.14, 1.46±0.19, 1.30±0.10 and 1.26±0.21 for gridboxes 3, 4, 5 and 6, respectively. Gridboxes farther north in the 2016 Meridional1 transect also were more humid, with f(RH) for gridboxes 1, 2 and 3 of 1.36±0.14, 1.39±0.14 and 1.44±0.51, respectively, in the 2-5km altitude range. If γ is actually lower than derived here – e.g., closer to a value of ~0.3, as measured in the year 2000 SAFARI campaign in southwest Africa (Titos et al., 2016) – the f(RH) values applied here would be about 30% too large (~1.5 vs. ~1.2 for γ of 0.65 vs. 0.3, for an ambient RH of 60%)."*

Line 1173. Having subsections for each model would be beneficial for the reader.

*Sub-sections have been added for each model within the section discussing the extinction comparisons, as suggested.*

Line 1173. The WRF-CAM5 section is difficult to follow, but the other models are better. I suggest the authors look at this model section and try to improve the clarity of it.

*The sub-sections discussing the results from the other models have been edited for better clarity.*

Figure 10. It would be useful to have a title above both columns to easily differentiate the two without having to refer to the caption.

*Apologies. The title was there but we did not check page alignment carefully enough so the title landed at the bottom of the previous page. The figures have overall been re-done for better clarity. The "in-situ" vs "HSRL" is now given as part of the x-axis label.*

Line 1224. 'closer to or greater than 1.0' this is not consistent though.

*We're not sure what the reviewer means by "this is not consistent". In some cases the ratio goes from <1.0 to something still <1.0 but closer to 1.0, and in some cases (e.g. in panel a), in the 2-3km altitude bin the mean ratio is slightly greater than 1.0).*

Figure 11. 'as in figure 8, but for the GEOS model' I don't think the cross ref is correct. There are also other instances of incorrectly cross-referenced figures so please check all captions.

*Apologies. References to figures / tables have been carefully checked with the re-do (and re-numbering) of the figures.*

Line 1520. 'SSA is consistently higher than that observed' but aren't observations for low RH? If so, isn't 'consistently higher than observed' actually good?

*That depends on whether SSA actually goes up with humidification, and by how much. The text notes that, in the models, SSA does systematically increase with RH. The text now also reads: "In contrast, for the two 2016 transects the ALADIN ambient-RH SSA is almost always higher than the observed dry SSA, with typical differences of 0.02-0.04. Adjustment of the ALADIN (ambient-RH) values to low RH (as in the observations) should bring the two into better agreement, though perhaps not for all transects: the differences are smaller in the 2017 Meridional2 transect, particularly towards the south, despite the higher ambient RH."*

Line 1523. What trends are being referred to here?

*Sentence has been edited for better clarity. It now reads: "SSA in ALADIN includes a dependence on aerosol aging and RH (Mallet et al., 2019), and thus it is not clear if this altitude dependence in SSA is a response to higher RH towards the top of the plume or if it would still be present under dry conditions."*

Figure 15. The observations bar makes it seem like you are comparing like for like, but it may be more appropriate to make the observations bar unfilled as it for low RH, and therefore more comparable to the UM-UKCA unfilled bar?

*Good point. The figure now shows the observed values with unfilled bars, as in the UM-UKCA "dry SSA" unfilled bar, and this is noted in the figure caption (now Figure 18).*

Line 1598. But the UM-UKCA dry vs ambient SSA are very different – doesn't this go against this statement? Does it suggest the model is completely wrong?

*It does indeed go against what the UM-UKCA model simulates. We have now added the following sentence: "They also imply that the impact of humidification on SSA in the UM-UKCA model (Figure 18) is too large (consistent with f(RH) being too large; Section 4.2.3.4), though it is difficult to make a robust conclusion based on this one observational comparison."*

Line 1614. Any ideas why this is occurring?

*We're also puzzled by this. The text has been edited to (hopefully) more clearly state what we know. We also note that resolving this would require work outside of the scope of this paper that considers more carefully the simulated size distribution and size-dependent composition of the aerosol.*

Line 1713. So the models are accidentally correct because they don't include absorption by brown carbon. Do the models provide information on brown carbon content?

*The text has been edited to clarify this, so it now reads:*
*"These four models simulate both black and organic carbon, but the organic carbon is not light-absorbing so AAE values are expected to be near 1. They do include the impacts of the addition of water on the aerosol indices of refraction and aerosol size which can drive variations in AAE from black carbon alone (i.e. of about 0.8-1.4; Liu et al., 2018)."*

Line 1965 (and 1988). Do you mean QFED2 rather than GFED?

*Good catch! Yes. Corrected, and thank you!*

Line 1972. Could differences in model dynamics lead to discrepancies?

*This question actually got us thinking more about the factors that could be contributing to the CO:BC ratio differences. Dynamics could be contributing through its impact on scavenging rates of BC. In addition, we realized that differences in "background" CO concentrations and in anthropogenic emissions could also be having some impacts. As such, the text has been edited to now read:*

*"In the core of the plume, low biases in BC in WRF-CAM5 and GEOS were somewhat smaller than the CO low biases. In other words, the CO:BC ratio was somewhat lower in the models than the observations, particularly in the October 2018 Meridional2 transect, and more so in the GEOS model than in WRF-CAM5. The ratio of CO:BC in biomass burning primary emissions depends on the material being burned and the efficiency of burning (smoldering versus flaming) (e.g. Reid et al., 2005), but this is set within the QFED emissions and so cannot explain these inter-model differences. Differences in anthropogenic sources could also be contributing (e.g. WRF-CAM5 uses EDGAR-HTAP while GEOS uses AeroCom Phase II), as could differences in the "background" CO concentrations. In October 2018, when differences between the models and the observations were largest, the biomass burning plume was less intense, so for that month the background and anthropogenic emissions could be more strongly influencing the CO and BC concentrations. For the biomass burning aerosol itself, given the common emissions dataset use by the two models, differences in the CO:BC ratio between the models must be due to differences in-atmosphere chemistry*

*and processing, combined with dynamics, leading to different aerosol scavenging rates. Notably, WRF-CAM5 has more sophisticated aerosol chemistry than GEOS. Greater scavenging losses in the first couple of days after emission, e.g., mostly over the land, in reality than in the models could also be contributing to the lower CO:BC ratio in models than in the observations.*

Line 2017. Has this been reported in previous literature?

*In brief, yes. The text now reads:*

*"In this case, a more likely source of the apparent low bias in model mass extinction efficiency are the real indices of refraction used in the models for aerosol components, as the OA real refractive index used in WRF-CAM5 is 1.45 while literature values for biomass burning are more often in the range 1.52-1.55 range (Aldhaif et al., 2018)."*

Line 2021. 'the plume top terminating at lower altitude in the observations than in some of the simulations' I thought the HSRL data showed that this wasn't actually the case and that the aerosol was indeed present at these higher altitudes? (see line 1236)

*Good point. We have added the following text: "It isn't fully clear, however, whether this is a robust result, given that the HSRL-2, like the models, measured extinction extending to higher altitudes than did the in-situ observations (Section 4.2.3)."*

Line 2093. '8-11 for all four' is this the range or difference? Please can you clarify

*Text now specifies this is the range.*

Line 2008. This estimation of the direct effect must implicitly assume that there are no rapid adjustments that may have influenced the underlying cloud field. Do you assume that all retrievals used are consistent with a sufficiently separated smoke-cloud scene? I think adding a sentence to clarify the assumption would be useful.

*We believe this comment is actually referring to line 2108, where the DARE calculation is defined. We have now added the following text at the end of the paragraph where the DARE calculation (Eqn [3]) is defined:*

*"This DARE estimate does not account for the fact that the cloud fields (and therefore $a\_s$) might have been affected by rapid adjustments to the smoke plume direct forcing (the "semi-direct" effect). Depending on the amount and altitude of heating by aerosol absorption above the clouds, this could have increased or decreased $a\_s$, thereby affecting the calculation of DARE using Eqn. [3]."*

**Comment on acp-2021-333**

Anonymous Referee #2

This comprehensive study utilizes aircraft observations from the NASA ORACLES campaign during three biomass burning seasons to analyze differences in modeled properties of aerosol plumes over the Southeast Atlantic. The modeling comparison to in situ observations was conducted using two regional models and two global models for the same temporal periods and specified aircraft transects. The work further extends insights into the importance of adequately modeling aerosol, cloud, and optical properties by demonstrating the propagation of biases in parameterized and simulated direct aerosol radiative effect (DARE) which leads to a range of largely positive and marginally negative values. Approaches and findings presented here are compelling and point to modeled parameters that require tuning for improved simulation of biomass burning aerosol on regional and global climate. This manuscript is overall well-written, though ordering and structure of the results sections contribute to some lack of clarity in the manuscript. Nevertheless, this work is worthy of publication in ACP if the results can be presented in such a way that they support the conclusions.

*The authors thank this reviewer for taking the time to read and provide a review for what is, we fully acknowledge, a long and involved paper. We hope that the resulting modifications address their concerns.*

**General Comments**

The problem with the presentation of these results is that the authors have not provided calculations with quantitative results or consolidated figures that support their main findings (plumes too diffuse, underestimates in plume properties, COT well simulated). The result is a meandering collection of multi-page graphs that the reader is expected to visually integrate and compare in order to reach the cryptic qualitative assessments described in the results. The authors need to provide summary figures that support their summary statements, preferably with numbers to describe them rather than qualitative descriptors like "low" and "high". I am not disputing any of these results, but I find it a disservice to readers to not provide them with figures that actually support, in a condensed way, the very generalized conclusions reached. Almost all of the existing figure panels belong in the SI, as they are not discussed in the text.

*We're not sure how to reply to the comment that quantitative results weren't calculated, as we give just that in Tables 1 and 2, as well as the figures. The simulated plumes being too vertically diffuse is shown in the figures, and the impacts (in some cases combined with vertical displacement of the plume in the model) is discussed quantitatively in the discussion. Similarly, overall mean biases are quantified for each 1km altitude bin, and for the 2D cloud properties – and all of this is used to quantify how biases in all modeled properties and each individual property lead to biases in DARE.*

*The figures in the main text showing the results for the BC, OA and extinction comparisons have been simplified and condensed, so that they now show only the transect-mean biases, and show results for all models in one panel. The figures originally in the main text, that also showed biases for each comparison gridbox, have been moved to the SI.*

The next big concern for this submission is the presentation of DARE as one of the leading messages of the manuscript. The section as a whole is well-written and findings sound, though its current position is awkward following the summary of the modeling comparisons and reads as an addendum to the manuscript. Given the combined impact of aerosol-cloud-optical properties on DARE (the main assessment of this work), a more appropriate placement for this major section would follow the discussion of cloud fraction and optical thickness biases (Section 4.4) and before the summary and concluding remarks (Section 5), which is neither a summary (as it is long and winding) nor concluding (as it is followed by DARE). Many passages within Section 4.4 and Section 5 repetitively allude to expected findings that are provided in the DARE section (Section 6) and add to the already exhaustive length of the manuscript. Removing these passages or placing them within context of a reordered DARE section would improve the flow from significance of aerosol-cloud-optical properties to climate impacts.

*The section on the DARE analysis has been moved to Section 5 and this is now followed by the Section 6 "Discussion and conclusions" section (renamed from "summary and conclusions" to "discussion and conclusions" since this section does discuss the findings and put them into a larger context, rather than simply "summarizing" them).*

**Minor Corrections:**

Line 102-113: Other than the supplementary modeling/forecasting information provided by UM-UKCA and ALADIN during their respective SE Atlantic campaigns, are there further reasons for using these models? A single regional and global model comparison to observations is a considerable effort. Are these additional models, which in some cases lack some comparison necessary variable fields, used only to expand the comparison discussion? A brief reference to this methodology choice would provide better perspective and support for the length of this manuscript.

*As noted in the text, the UM-UKCA and ALADIN models were used in the other two field campaigns that took place in parallel with ORACLES: the UK CLARIFY project and the French AeroClo-SA project, respectively. While they did not have as many of the comparison variables available, it seemed opportune to provide comparisons to the variables that were available. This hopefully will allow better interpretation of results under all three projects, providing a more coherent view of the biomass burning plume and it's impacts over the SE Atlantic region.*

Line 250: Provide the typical size range of accumulation mode aerosol, particularly as it pertains to biomass burning aerosol. Redemann et al. (2021)[1] provides support of this claim.

*The text now reads: "Above the boundary layer, the aerosol during ORACLES was dominated by accumulation mode biomass burning smoke with a volumetric mean diameter of <0.4µm (e.g. see Figure 8 of Shinozuka et al., 2020)."*

Figure 3: Average symbols (black circles) are not clearly identifiable in these panels. Symbols with a bolder weight would improve the presentation and clarity of these figures. This should also be addressed in Figures 7-13 and subsequent supplementary figures (e.g. Figure S.4). This figure (and several others) was paginated as 3 separate pages, making it pretty unwieldy to review as a single figure. The authors should consider a format with less wasted space and higher density of information. Also, the scatter plot approach makes it challenging to see vertical trends for each color; I recommend adding a connecting line.

*This figure has been broken down into three separate figures and the symbol size increased. Each figure now fits onto a page. Vertical lines were not added as this makes the figures even more difficult to view.*

Fig. 4-14: as with Fig. 3, please rearrange, shrink, or break up into figures that each fit on a single page, with relevant axis labels and legends on the same page and provide captions written upright.

*The figures have been altered to now only show the transect-mean biases in the main article figures, with individual gridbox biases given in supplemental figures. The symbols used in the resulting plots now match those shown in the figures showing e.g. profiles of extinction and are easier to read. The figures also are not separated to show fewer panels per figure, generally grouping 2016 results and 2017+2018 results together. All figures and associated captions are now in portrait orientation and each fits within a page.*

Figure 15: This figure seems superfluous in the main text of the manuscript given the inclusion of vertical profiles of SSA in Figure 14. The statistical inter-model comparison of SSA is interesting, but may be better suited for the supplementary text.

*We disagree; we think the histograms give a better overall sense of the significant differences in SSA across the models and observations. The vertical profiles then show the vertical dependence of the biases. We will leave it up to the editor to decide if this figure should be cut.*

Line 658 "Future studies may want to use a synthesis of modeled and satellite-retrieved properties (e.g. of AOD) for a more robust analysis of sampling representativeness." Is there a reason this is not done here, i.e. at a minimum a comparison to the avg+SD of the models shown? If the models are too disparate to make this meaningful, then why use them?

*The decision on what statistics would be produced by the models for this comparison study were made early on in the ORACLES project. Only two of the models (WRF-CAM5 and GEOS) produced the statistics to test for representativeness. It was only after running the statistics for this paper that we found the inter-model differences in whether the sampled plume was representative of the month-long average. At this point it was too late in the project to request new variables or model runs. This lesson will have to be left to inform future studies.*

*In any case, this text has now been deleted as we have added to Section 2.2 discussion of relative variability in extinction in the models versus in observations, as a metric for how robust a test of representativeness the models provide. The text above has now also been replaced with reference to this.*

Line 967: "UM-UKCA has a significant low bias in BC at plume altitudes in 2016 and a smaller low bias in 2017 (Figure 9)." This statement (and many others like it) are difficult to support from the figure noted (9). This reflects two problems (1) There are no multi- flight-type results presented from which to infer the yearly differences noted, suggesting perhaps that one should be able to eyeball this to see and quantify the low bias? And (2) There is no metric or test presented for significance of the 2016 result vs. 2017. Both of these issues are present throughout the manuscript, with figures showing individual results but text describing trends amalgamated over several graphs.

Of course there are many ways to consolidate these results (by region, by altitude, etc.), and the way chosen will of necessity be limited to the particular aim of this work. But the failure to present any such consolidation results in a mismatch between the text and the figures. A much more useful paper would present the consolidated papers and move the detailed figures and tables to the SI.

*The text has been edited for better clarity. Most importantly, we note that these biases are quantified, in Table 1. We now point to Table 1.*

*We are not sure what is meant by "there are not multi-flight-type results presented from which to infer the yearly differences noted". That is, in fact, exactly what this comparison does: It consolidates multiple flights within each year and comparison transect to get statistics on the ratio of modeled-to-observed aerosol properties.*

*Figures in the main paper showing ratios of modeled to observed BC mass, OA mass and $\sigma_{ep}$ have been simplified to show only the transect-mean biases at each altitude. The more detailed plots have been moved to Supplemental figures.*

Line 1123: This is super interesting; how does it compare to past comparisons, e.g. Heald et al. 2008(?) or similar?

*We assume the reviewer is referring to:*

*Heald, C. L., Goldstein, A. H., Allan, J. D., Aiken, A. C., Apel, E., Atlas, E. et al.: Total observed organic carbon (TOOC) in the atmosphere: a synthesis of North American observations, Atmos. Chem. Phys., 8, 2007–2025, https://doi.org/10.5194/acp-8-2007-2008, 2008.*

*The findings in that paper support the assertion that while OA can be produced in the ocean it is still a small fraction of the accumulation-mode aerosol, so we have added reference to this paper to the text.*

Line 1136: Correct "humification" to "humidification".

*Thank you for this; it has been corrected.*

Line 1233: "Except in the 2018 Meridional2 transect, WRF-CAM5 $\sigma_{ep}$ is generally 30-40% lower than measured by HSRL-2." As l.967, where is this shown?

*This is shown in Table 2. Table 2 is now referenced.*

Line 1309: "The bias also has a less consistent dependence on altitude in 2017 (Figure 11). In 2018, the higher ambient RH could be

*This comment is incomplete so we are unable to address it.*

1310 compensating for some of the low bias in dry aerosol $\sigma_{ep}$."As l.967, where is this shown?

*The higher RH is shown in Figure 6. This is now referenced.*

1320 "In the 2016 Diagonal transect this produces significant low model biases for 3-5 km altitude and high biases for 2-3 km and 5-6 km (Figure 12; Table 1), much" What is "significant low" and what is "high"? Which values or plots are referenced?

*This sentence has been corrected to read: "In the 2016 Diagonal transect this produces a low biases in modeled $\sigma_{ep}$ that increases with altitude from 3km to 5 km, and high biases below 3 km (Figure 14a; Table 1), much as for the GEOS model." The actual numerical biases are given in the table and can be viewed in the figure.*

Lines 1601-1603: This sentence is hard to follow and should be revisited for clarity. Are the authors trying to say a larger number of small particles less than 1000 nm dry diameter as is the case in the biomass burning plume?

*This sentence has been rewritten to read: "Whereas SSA varies primarily with aerosol composition, SAE varies primarily with aerosol size. For aerosol smaller than approximately 1000 nm dry diameter – i.e. as for aerosol in the SE Atlantic biomass burning plume (Shinozuka et al., 2020) – SAE gets smaller as aerosol size increases (Schuster et al., 2006)."*

Line 2055: Remove "the"

*Thank you for the catch. Done.*

Lines 2201-2202: Suggest "to produce values of DAREavg that is a factor of" to "to produce a DAREavg that is a factor of".

*Done.*

**References:**

[1]https://acp.copernicus.org/articles/21/1507/2021/